# Nivolumab and ipilimumab in recurrent or refractory cancer of unknown primary: a phase II trial

Maria Pouyiourou [1,2,3,17], Bianca N. Kraft[1,2,17], Timothy Wohlfromm[1], Michael Stahl[4], Boris Kubuschok [5], Harald Löffler[6], Ulrich T. Hacker[7], Gerdt Hübner[8], Lena Weiss[9], Michael Bitzer [10], Thomas Ernst [11], Philipp Schütt[12], Thomas Hielscher [13], Stefan Delorme [14], Martina Kirchner[15,16], Daniel Kazdal[15,16], Markus Ball [15,16], Klaus Kluck [15,16], Albrecht Stenzinger [15,16], Tilmann Bochtler[1,2,3] & Alwin Krämer [1,2,3] ✉

Cancer of unknown primary has a dismal prognosis, especially following failure of platinum-based chemotherapy. 10-20% of patients have a high tumor mutational burden (TMB), which predicts response to immunotherapy in many cancer types. In this prospective, non-randomized, open-label, multicenter Phase II trial (EudraCT 2018-004562-33; NCT04131621), patients relapsed or refractory after platinum-based chemotherapy received nivolumab and ipilimumab following TMB[high] vs. TMB[low] stratification. Progression-free survival (PFS) represented the primary endpoint; overall survival (OS), response rates, duration of clinical benefit and safety were the secondary endpoints. The trial was prematurely terminated in March 2021 before reaching the preplanned sample size ($n = 194$). Among 31 evaluable patients, 16% had a high TMB ( > 12 mutations/Mb). Overall response rate was 16% (95% CI 6-34%), with 7.7% (95% CI 1-25%) vs. 60% (95% CI 15-95%) in TMB[low] and TMB[high], respectively. Although the primary endpoint was not met, high TMB was associated with better median PFS (18.3 vs. 2.4 months) and OS (18.3 vs. 3.6 months). Severe immune-related adverse events were reported in 29% of cases. Assessing on-treatment dynamics of circulating tumor DNA using combined targeted hotspot mutation and shallow whole genome sequencing as part of a predefined exploratory analysis identified patients benefiting from immunotherapy irrespective of initial radiologic response.

Cancer of unknown primary (CUP) is a heterogenous aggressive malignancy characterized by histologically confirmed metastatic spread in the absence of a primary tumor responsible for metastatic seeding[1]. Based on clinical and histological criteria, CUP can be classified into favorable and unfavorable subgroups, the latter typically demonstrating extensive metastatic dissemination and a median overall survival (OS) of only 3–12 months. Standard of care in

unfavorable CUP typically comprises platinum-based doublet chemotherapy as first-line treatment[1–10]. Further-line chemotherapy is not established and leads to a median PFS and OS of only 2–4 and 3–9 months, respectively[11–16], creating a high unmet medical need for new therapeutic approaches.

Immune checkpoint inhibitors (ICI) have evolved into indispensable components of the standard therapy for several cancer

---

entities. However, since only a minority of patients derive durable benefit from ICI treatment, the need for robust predictors of response is immense. Tumor mutational burden (TMB) has emerged as an independent predictive biomarker of ICI efficacy. Based on the results of the KEYNOTE-158 study linking high TMB (TMB[high]) to better response and survival to immune checkpoint blockade across several solid tumor types, the Food and Drug Administration (FDA) granted approval for the use of pembrolizumab in TMB[high] patients regardless of the cancer´s tissue of origin, thereby including CUP[17]. Several studies analyzing the genomic profile of CUP reported that about 10–20% of cases harbor high TMB levels[18–21]. In order to determine the value of immunotherapy in CUP, prospective data on treatment efficacy and biomarker reliability are needed.

Given the aggressive nature of unfavorable CUP, reliable early-on monitoring of therapeutic response is instrumental. The noninvasive longitudinal analysis of circulating cell free DNA (ccfDNA) and its tumor-derived fraction (ctDNA) has gained significant attention for this purpose, with its clinical utility currently being investigated in several tumor entities[22–25].

We here report the results of the CheCUP trial, a multicenter phase II study of combined nivolumab (PD-1 checkpoint inhibitor) and ipilimumab (CTLA-4 checkpoint inhibitor) in patients with unfavorable CUP relapsed after or refractory to platinum-based chemotherapy (NCT04131621, EudraCT No. 2018-004562-33). Combined ultra-deep targeted next-generation sequencing (NGS) of patient-specific hotspot mutations and copy number alteration (CNA) profiling by low-coverage, shallow whole genome sequencing (sWGS) of ccfDNA from serial plasma samples was used to evaluate response to ICI treatment and compared to radiologic response assessment and survival. We find that 60% of CUP patients with high TMB respond to combined ICI therapy, with early on-treatment ctDNA analysis reliably identifying patients deriving long-term treatment benefit.

## Results

### Study design and baseline patient characteristics

Between December 2019 and March 2021, a total of 45 patients with unfavorable CUP, who were relapsed after or refractory to at least three cycles of platinum-based first-line chemotherapy were screened for study inclusion (Fig. 1a, b). Fourteen patients were excluded: three patients had no tumor samples available for TMB assessment, nine patients did not meet the trial inclusion criteria, one patient withdrew consent, and one patient was diagnosed with intrahepatic cholangiocarcinoma instead of CUP as a result of the review of CT imaging by the reference radiologist of the trial. The trial was terminated early by the sponsor due to insufficient patient enrollment. As a result, 31 CUP patients were enrolled into the trial, stratified as either TMB[high] or TMB[low] based on a TMB cut-off of 12 mutations/Mb and received combined nivolumab (240 mg biweekly) and ipilimumab (1 mg/kg every 6 weeks) until disease progression, unacceptable toxicity or death from any cause (Fig. 1b). Progression-free survival (PFS) was the primary endpoint and overall survival (OS), overall response rate (ORR), duration of clinical benefit and safety objectives were predefined secondary endpoints. Due to the limited sample size, no confirmatory testing was performed. Instead, all analyses were exploratory and P values are to be interpreted accordingly.

The trial population reflected a typical CUP cohort in terms of histology, demographical and clinical characteristics (Table 1). In total, 64.5% of patients had adenocarcinoma, 16.1% squamous cell carcinoma and 12.9% undifferentiated carcinoma, while carcinomas with sarcomatoid differentiation were reported in 6.5% of cases. Following targeted NGS of CUP tissue, five patients were stratified as TMB[high] (median TMB 18.91, range 13.3–22.8 mutations/Mb) and 26 patients as TMB[low] (median TMB 4.7, range 0–7.84 mutations/Mb).

Patients had been previously treated with a median of two therapy lines (range 1–5), including platinum-based chemotherapy (39%

carboplatin, 39% cisplatin, 34% oxaliplatin) as defined by the inclusion criteria. Ten patients (32.3%) had previously undergone radiotherapy to treat bone (16.1%), liver (3.2%) or lymph node metastases (9.7%). To better reflect the disease burden of the study patients, we developed a metastasis burden score, which, in addition to target lesion diameters, includes the numbers of affected organs and metastases per organ (Supplementary Fig. 1), as these parameters have been shown to impact on CUP patient prognosis[5,26]. Metastasis burden score calculation and patient categorization criteria are detailed in Supplementary Table 1. While 73% of the TMB[low] patients suffered from an intermediate or high metastatic burden, only 40% of the TMB[high] patients were classified into these categories (Table 1). The distribution of clinical parameters between the TMB[high] and TMB[low] strata is summarized in Table 1.

PD-L1 expression status was available for 15 patients: eight patients (25.8%) had a combined positive score (CPS) and/or tumor proportion score (TPS) of at least 1% and were considered PD-L1-positive, while seven patients (22.6%) were PD-L1-negative according to these criteria (Supplementary Table 2). Only two patients had CPS and/or TPS ≥ 50%.

### Predictive value of TMB for ICI treatment response and survival

The median follow-up duration for PFS and OS at data cut-off (March 15, 2022) was 7.6 and 14.9 months, respectively. 25 PFS and 24 OS events were documented. Among all included patients, the one-year PFS and OS rates were 18% and 26%, respectively. Median PFS was 2.5 months (95% confidence interval [CI] 1.77–3.32), while the median OS was 3.8 months (95% CI 3.3–8.8) (Supplementary Fig. 2). TMB[high] patients received a median of nine ICI treatment cycles (range 1–18) as compared to two cycles (range 1–16) in the TMB[low] group. Three patients were still on treatment at data cut-off.

Dual checkpoint blockade led to two complete (6.5%) and three partial remissions (9.7%) according to RECIST v1.1, summing up to an ORR of 16.2% (95% CI 5.5-33.7%). Among these, three of five patients (60%; 95% CI 14.7–94.7%) with high TMB and two of 26 patients (7.7%; 95% CI 0.9-25.1%) with low TMB achieved an objective response (Fisher´s exact test $p = 0.02$; Table 2, Supplementary Table 3). Median time to response was 80 days (range 29-161). Duration of clinical benefit ranged between 53 and 101 weeks, with three of five objective responses going on at database lock. One patient who had previously achieved a partial remission progressed after 21 months of treatment and a second patient died of sepsis while in complete remission 18 months after treatment start. Stable and progressive disease was found in one (3.2%) and twelve cases (38.7%), respectively. In thirteen additional patients (41.9%) study medication had to be terminated prior to first response assessment. Reasons included intolerable adverse events in two cases (15.4%) and clinically suspected disease progression in eight patients (61.5%), of which only one belonged to the TMB[high] group. Upon progression four patients received subsequent treatment, which consisted of chemotherapy in three and targeted therapy with ivosidenib in one case with activating IDH1-p.R132L mutation. Histology and immunohistochemistry of this case were compatible with a primary tumor in the upper gastrointestinal tract but not conclusive for cholangiocarcinoma. Reference radiology at study eligibility screening did not confirm the diagnosis of cholangiocarcinoma according to the differential diagnostic algorithms of the ESMO CUP guidelines[1]. None of these patients achieved a sustained response with post-immunotherapy treatment. Disease progression was the main cause of death during follow-up (20 of 24 cases, 83.3%).

No treatment-related mortality occurred. Safety data are shown in Table 3. The most common adverse events were infections, followed by nausea and renal impairment (32.2%, 19.4% and 19.4%, respectively). Eighteen serious adverse events were reported. Treatment-related adverse events led to withdrawal of ipilimumab ($n = 1$; 3.2%) or both

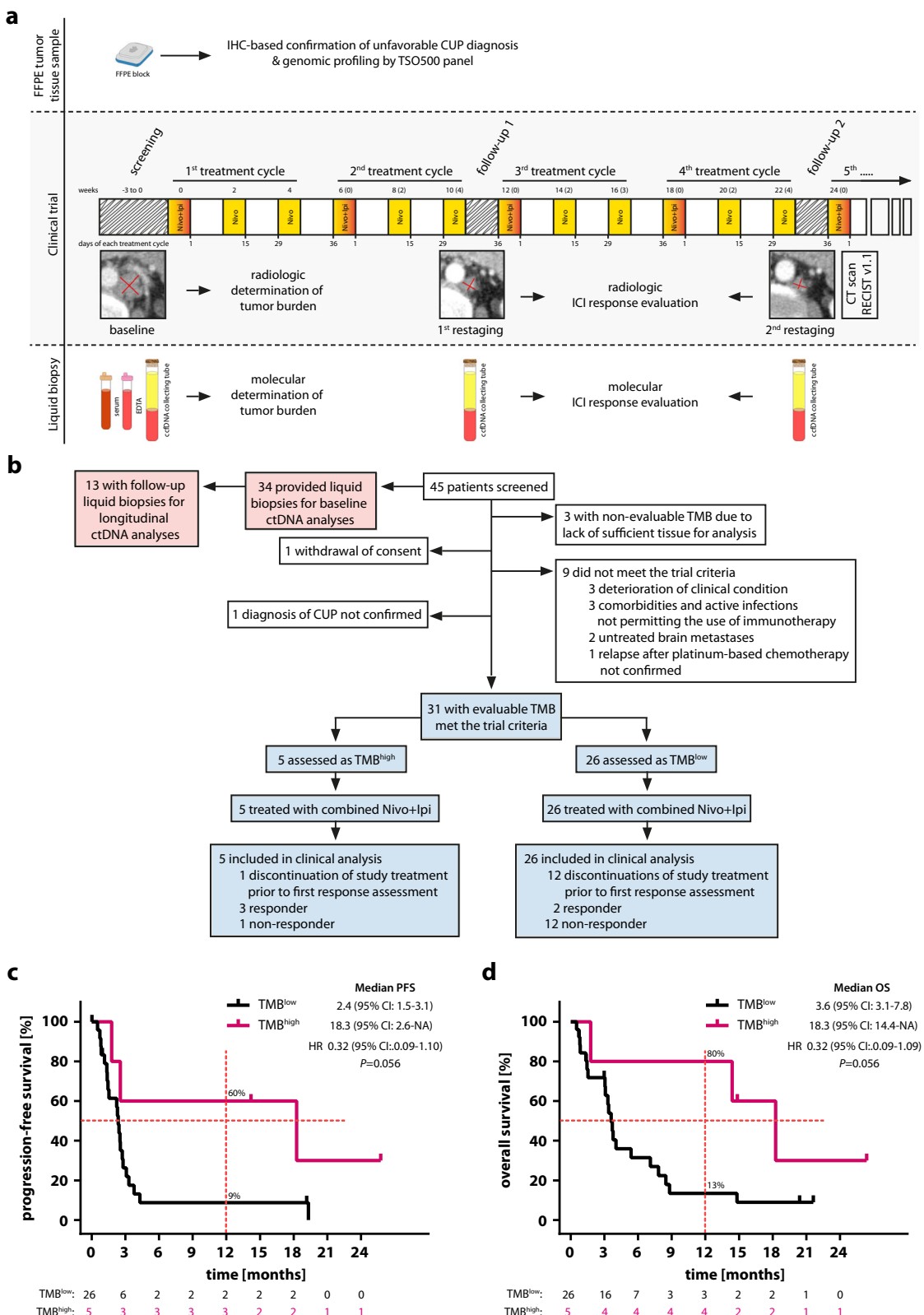

study drugs (*n* = 3; 9.7%) in four cases, while treatment was temporarily discontinued due to adverse events in twelve patients. One patient died of neutropenic sepsis during the thirteenth treatment cycle, probably as a result of preexisting chemotherapy-induced bone marrow insufficiency. The rates of immune-related adverse events were consistent with previous observations[27].

Survival analysis in patients with high vs. low TMB revealed a clinically meaningful longer PFS in the TMB$^{high}$ group. The one-year PFS rate of the TMB$^{high}$ group was 60% compared to 9% in the TMB$^{low}$ cohort, and median PFS was 18.3 versus 2.4 months, respectively (log-rank *p* = 0.056, HR 0.32, 95% CI 0.09–1.10, Fig. 1c). One-year OS rates of TMB$^{high}$ and TMB$^{low}$ cohorts were 80% and 13%, respectively. Median OS

**Fig. 1 | High TMB is predictive of clinical outcome after combined administration of nivolumab and ipilimumab in patients with recurrent or refractory unfavorable CUP. a** CheCUP trial design. Patients with recurrent or refractory unfavorable CUP provided tumor samples to specify TMB by comprehensive genomic profiling and underwent baseline CT scan of the neck, chest and abdomen. Patients enrolled in the CheCUP trial had at least one measurable lesion according to RECIST v1.1. All patients who met study criteria received nivolumab (240 mg) every two weeks and ipilimumab (1 mg/kg) every six weeks until disease progression. Response to ICI treatment was evaluated by the trial radiologists according to RECIST v1.1 at follow-up visits every second treatment cycle. If patients consented to translational research, liquid biopsy samples (serum, whole blood for PBMNCs and plasma isolation) suitable for ctDNA analyses were collected in parallel with radiological assessment. **b** Schematic outlining of patient enrollment and specimens involved in this study. The CheCUP trial cohort (*blue*) was stratified as either TMB^high or TMB^low based on a TMB cut-off of 12 mutations/Mb following comprehensive genomic profiling of tumor tissue. The translational research cohort (*red*) provided liquid biopsy samples for on-treatment ctDNA analyses. Kaplan-Meier estimates of (**c**) PFS and (**d**) OS, stratified according to patient's TMB status: TMB^high (*n* = 5) and TMB^low (*n* = 26). Crosses denote censored observations, and for each time interval the number of patients at risk are indicated below the plots. Comparisons were made using a two-sided log-rank test, Cox proportional hazard regression modeling was used to calculate hazard ratio. The horizontal dashed lines mark the median values, the vertical dashed lines the one-year values. 95% CI, 95% confidence interval; HR, hazard ratio. Source data are provided as a Source Data file.

was 18.3 months in TMB^high versus 3.6 months in TMB^low patients (log-rank $p = 0.056$, HR 0.32, 95% CI 0.09–1.09; Fig. 1d).

Both PFS and OS were significantly longer in patients with low compared to patients with intermediate or high metastasis burden scores (median PFS 18.3 vs. 1.45 vs. 2.3 months, log-rank $p = 0.005$; median OS 18.3 vs. 3.1 vs. 3.6 months, log-rank $p = 0.002$; Fig. 2a, b). Likewise, higher ECOG status was associated with increasing risk for progression and death (median PFS 2.7 vs. 2.4 vs. 0.9 months, log-rank $p = 0.005$; median OS 11.1 vs. 3.3 vs. 0.9 months, log-rank $p = 0.003$; Fig. 2c, d). In contrast, survival was not affected by the number of previous chemotherapy lines (Fig. 2e, f). Squamous cell histology was associated with a trend towards superior OS without reaching statistical significance (Supplementary Fig. 3a, b), while PD-L1 expression led to a trend towards favorable OS but not PFS (median OS 11.8 vs. 3.7 months, log-rank $p = 0.09$; median PFS 2.7 vs. 2.6 months, log-rank $p = 0.372$; Supplementary Fig. 3c, d). Adjusting for PD-L1 expression by multivariate analysis did not affect the impact of TMB on OS (HR 0.31, 95% CI 0.06–1.58, $p = 0.16$) but decreased its impact on PFS (HR 0.49, 95% CI 0.10–2.46; $p = 0.39$ Table 4). Upon adjusting for ECOG and metastasis burden score, the favorable effect of TMB on OS only slightly decreased (HR 0.45, 95% CI 0.11–1.79; $p = 0.26$), while the effect on PFS was attenuated again (HR 0.70, 95% CI 0.18–2.67; $p = 0.60$; Table 4). After multivariate analysis, the effect of TMB on OS ($p = 0.26$) and PFS ($p = 0.60$) was not statistically significant.

Based on clinical and immunohistochemical features patients were independently classified for their putative primary by three experienced oncologists as previously described[21], with putative primary tumors being only registered in case of consensus between at least two of the three investigators. As a result, 16 of 31 cases were assigned to either lung (6 cases, 19.4%), upper gastrointestinal (5 cases, 16.1%), anal/cervix (3 cases, 9.7%), breast (one case, 3.2%) or colon (one case, 3.2%) cancer, while 13 cases (41.9%) were considered fully enigmatic, and two cases could not be unanimously assigned to any of these groups. Upon comparing treatment efficacy according to putative primary sites, no statistically significant differences in treatment response or survival were detected (Supplementary Fig. 4).

### Identification of additional ICI response predictors by genomic analysis of pretreatment CUP metastasis tissue

Panel sequencing data from baseline FFPE tumor biopsies were available for 29 of 31 patients. None of the samples showed microsatellite instability (MSI). Panel sequencing of 521 cancer-relevant genes revealed at least one molecular alteration in 28/29 CUP patients, including single nucleotide variants (SNVs) in 26 cases (89.7%) and CNAs, as revealed by sWGS, in 21 cases (72.4%) (Fig. 3a, Supplementary Fig. 5). Potentially clinically relevant mutations were detected in 58 genes (Supplementary Fig. 5a). One patient had a highly rearranged tumor genome, preventing unambiguous CNA analysis. CNAs of the remaining patients ranged from single gene deletions/gains to large chromosome region losses/gains. In 29 CUP tumor genomes, a total of 266 gene deletions and 203 gene gains were detected among the 521 genes analyzed (Supplementary Fig. 5b). Individual CUP cases harbored up to eight clinically relevant SNVs, 57 and 132 genes showing allele gains or deletions, respectively, or 159 molecular alterations in total (Supplementary Fig. 6).

Almost all gene/chromosome alterations were unique, present in only one or two out of the 29 patients. Exceptions were alterations of *CDKN2A* (48.3%), *CDKN2B* (31.0%), *TP53* (37.9%) and *KRAS* (27.6%), consistent with previous findings[8,18,26,28,29], and corroborating that the study patients comprised a typical CUP cohort (Fig. 3a). Most (6/8, 25%) *KRAS* alterations consisted of an activating mutation at amino acid position p.G12. Two additional patients had activating *NRAS* alterations. All eleven *TP53* mutations were private, present only in one individual, and functionally deleterious. Moreover, two patients showed *TP53* loss of heterozygosity (LOH) (Fig. 3a). In contrast to non-small cell lung cancer, where *RAS* and/or *TP53* mutations increase sensitivity to PD-L1 blockade[30–33], CUP patients with *RAS* and/or *TP53* alterations did not benefit from ICI treatment. To the contrary, patients with activated *RAS* signaling or deleterious *TP53* mutations exhibited significantly shorter PFS (*RAS*, log-rank $p = 0.066$, HR 2.23, 95% CI 0.92-5.41; *TP53*, log-rank $p = 0.036$, HR 2.39, 95% CI 1.03–5.55) and OS (*RAS*, log-rank $p = 0.007$, HR 3.43, 95% CI 1.33-8.88; *TP53*, log-rank $p = 0.043$, HR 2.34, 95% CI 1.00–5.43) than patients without these alterations (Fig. 3b, c). In addition, DNA damage repair pathway gene alterations predictive for PD-L1 blockade sensitivity in several tumor types[34–36] did not show a statistically significant beneficial effect on survival either (Supplementary Fig. 7).

For homologous recombination deficiency (HRD) scoring, off-target reads were counted in 100 KB segments for genome-wide CNA profiling. HRD score estimates of the analyzed tumor samples ranged from 7 to 62 with a median of 22 (Table 1). While the majority of cases ($n = 19$, 66%) with HRD estimates below 30 were classified as HRD-negative, two samples had HRD estimates above 50, indicating HRD positivity. HRD positivity did not predict for superior PFS or OS (median PFS 2.4 vs 2.6 vs 1.4 months, log-rank $p = 0.2$; median OS 3.7 vs 7.5 vs 1.4 months, log-rank $p = 0.2$).

Most of the patients with *CDKN2A* alterations (12/14, 85.7%) harbored a whole-gene deletion of the tumor suppressor, which was in 8/12 cases (66.7%) accompanied by co-deletion of the adjacent *CDKN2B* locus (Fig. 3a). *CDKN2A* deletion was not associated with a significant reduction in survival (Supplementary Fig. 8a). Interestingly, in 5/8 (62.5%) patients with *CDKN2A/CDKN2B* co-deletion, the deleted region at chromosome 9p21.3 included a cluster of 16 genes encoding type I interferons (IFN I). Four of these five patients harbored an additional 9p24 region loss including *JAK2*, *CD274* (PD-L1) and *PDCD1LG2* (PD-L2) gene loci (Fig. 3a). Of note, tumor biopsies of two patients with PD-L1 gene deletion were nevertheless PD-L1-CPS-positive by immunohistochemistry. Inactivating mutations and whole-gene deletions of the tumor suppressor gene *STK11* were identified in six patients (20.7%). Two of these patients had a concurrent *KEAP1* mutation. Deletions of 9p21.3, 9p24, *STK11* and *KEAP1* are associated with an immunologically 'cold' tumor microenvironment and immunotherapy resistance[37–44]. As

## Table 1 | Baseline characteristics of the study population

| | Total | TMB^low | TMB^high |
|---|---|---|---|
| Patients (n) | 31 | 26 | 5 |
| **Age—yr** | | | |
| Median | 64 | 64 | 64 |
| Range | 33-77 | 33-77 | 44-70 |
| **Sex—no. (%)** | | | |
| Male | 15 (48.4) | 13 (50) | 2 (40) |
| Female | 16 (51.6) | 13 (50) | 3 (60) |
| **ECOG performance status—no. (%)** | | | |
| 0 | 12 (38.7) | 9 (34.6) | 3 (60) |
| 1 | 16 (51.6) | 14 (53.8) | 2 (40) |
| 2 | 3 (9.7) | 3 (11.5) | 0 |
| **Histology—no. (%)** | | | |
| Adenocarcinoma | 20 (64.5) | 18 (69.2) | 2 (40) |
| Poorly differentiated carcinoma | 4 (12.9) | 5 (15.4) | 0 |
| Squamous cell carcinoma | 5 (16.1) | 3 (11.5) | 2 (40) |
| Carcinoma with sarcomatoid differentiation | 2 (6.4) | 1 (3.8) | 1 (20) |
| **Number of affected organs—no. (%)** | | | |
| 1 | 8 (25.8) | 6 (23.1) | 2 (40) |
| 2 | 9 (29) | 8 (30.8) | 1 (20) |
| 3 | 9 (29) | 7 (26.9) | 2 (40) |
| 4 | 4 (12.9) | 4 (15.4) | 0 |
| 5 | 1 (3.2) | 1 (3.8) | 0 |
| **Metastatic sites—no. (%)** | | | |
| Liver | 16 (51.6) | 15 (57.7) | 1 (20) |
| Lung | 12 (38.7) | 10 (38.5) | 2 (40) |
| Bone | 8 (25.8) | 6 (23.1) | 2 (40) |
| Pleura | 3 (9.7) | 2 (7.7) | 1 (20) |
| Peritoneum | 5 (16.1) | 5 (19.2) | 0 |
| Lymph node | 19 (61.3) | 16 (61.5) | 3 (60) |
| Adrenal gland | 2 (6.5) | 2 (7.7) | 0 |
| Soft tissue | 7 (22.6) | 6 (23.1) | 1 (20) |
| Skin / subcutaneous tissue | 3 (9.7) | 3 (11.5) | 0 |
| Other | 6 (19.4) | 6 (23.1) | 0 |
| **Metastasis burden score—no. (%)** | | | |
| Low | 10 (32.3) | 7 (26.9) | 3 (60) |
| Intermediate | 10 (32.3) | 9 (34.6) | 1 (20) |
| High | 11 (35.5) | 10 (38.5) | 1 (20) |
| **Prior lines of chemotherapy—no. (%)** | | | |
| 1 | 14 (45.2) | 12 (46.2) | 2 (40) |
| 2 | 9 (29) | 6 (23.1) | 3 (60) |
| ≥ 3 | 8 (25.8) | 8 (30.8) | 0 |
| **Prior radiotherapy—no. (%)** | | | |
| Yes | 10 (32.3) | 9 (34.6) | 1 (20) |
| No | 21 (67.7) | 17 (65.4) | 4 (80) |
| **PD-L1 expression > 1%—no. (%)** | | | |
| Negative | 7/15 (46.7) | 6 (54.5) | 1 (25) |
| Positive | 8/15 (53.3) | 5 (45.5) | 3 (75) |
| **HRD score estimate** | | | |
| Negative | 19/29 (65.5) | 17 (70.8) | 2 (40) |
| Intermediate | 8/29 (27.6) | 5 (20.8) | 3 (60) |
| Positive | 2/29 (6.9) | 2 (8.3) | 0 |

*ECOG* Eastern Cooperative Oncology Group, *PD-L1* programmed death-ligand 1, *HRD* homologous recombination deficiency.
Source data are provided as a Source Data file.

## Table 2 | Treatment response

| | Total | TMB^low | TMB^high |
|---|---|---|---|
| Patients (n) | 31 | 26 | 5 |
| Overall response-n (%) | 5 (16.1) | 2 (7.7) | 3 (60) |
| **Best overall response-n (%)** | | | |
| Complete response | 2 (6.5) | 0 (0) | 2 (40) |
| Partial response | 3 (9.7) | 2 (7.7) | 1 (20) |
| Stable disease | 1 (3.2) | 1 (3.8) | 0 (0) |
| Progressive disease | 12 (38.7) | 11 (42.3) | 1 (20) |
| Early study discontinuation | 13 (41.9) | 12 (46.2) | 1 (20) |

Source data are provided as a Source Data file.

our genomic analysis revealed a high frequency of alterations associated with ICI resistance in previous studies (Supplementary Fig. 8b), we tested their impact on survival in our patient cohort. Statistically significant differences were neither observed for PFS nor OS in patients with ($n = 16$) and without ($n = 13$) ICI resistance-associated alterations (Supplementary Fig. 8c). To the contrary, both patients achieving complete remission on ICI treatment exhibited an ICI resistance-associated alteration (Fig. 3a).

To determine whether the immune cell composition of the tumor microenvironment predicts response to ICI therapy, gene expression profiles including 770 genes were generated from baseline FFPE tumor biopsies of 13 of the 31 patients using NanoString nCounter technology. The abundance of 14 specific immune cell populations was estimated from the mRNA expression profiles as recently described[45,46]. A significant correlation (false discovery rate <10%) with TMB was observed for neutrophils ($p = 0.0026$) and regulatory T (Treg) cells ($p = 0.011$, Supplementary Fig. 9). At the gene expression level, significant positive correlations with TMB were observed for 23 genes (*ZC3H12A, ENO1, IL1B*, SPP1, *IKBKG, CXCL8, FCAR, FOSL1, PSMB5, CD4, ITGB2, CSF3R, ITGAX, HK2, NOD2, SBNO2, SLC11A1, TFRC, ANLN, HK1, MYC, TYMP, TNFSF9*; $p < 0.0031$). Conversely, *RAD50* exhibited a significant negative correlation with TMB (p = 0.0015). No significant correlation with PFS or OS was found for any of the 14 immune cell populations analyzed (Supplementary Table 4).

In addition to TMB, a retrospective analysis has recently shown that tumor aneuploidy predicts survival following immunotherapy across multiple cancer types including CUP[47]. In this analysis, a high aneuploidy score was specifically associated with poor prognosis after ICI treatment among tumors with low but not high TMB. To determine the predictive value of aneuploidy for ICI treatment response and survival of CUP patients in a prospective manner, we have calculated aneuploidy scores, defined as the fraction of chromosome arms afflicted by arm-level CNAs from baseline FFPE tumor biopsies of 19/31 patients[48]. Of note, a higher aneuploidy score was associated with a significantly shorter PFS but not OS both in the total (Fig. 3d) and the TMB^low CheCUP population (Supplementary Fig. 10a). Moreover, 4 of 11 patients (36.4%) with low aneuploidy score, but none of eight patients with high aneuploidy score achieved an objective response.

### Longitudinal ctDNA analysis by combined targeted NGS and sWGS-based CNA profiling

To evaluate response to combined ICI therapy, we longitudinally collected plasma samples upon enrollment (baseline) and in three-monthly intervals (follow-up) to analyze ccfDNA and its tumor-derived ctDNA fraction (Fig. 1a) as part of a predefined exploratory analysis. Baseline plasma samples for ccfDNA isolation were available from 34 CUP patients, including 29/31 patients enrolled into the CheCUP trial and five additional CUP patients relapsed after or refractory to platinum-based standard first-line chemotherapy that were screened

**Table 3 | Incidence of treatment-related adverse events**

| Event | Cases (31 (%)) | |
|---|---|---|
| Any event | 27 (87.1) | |
| Any serious event | 18 (58.1) | |
| Any event leading to discontinuation or withdrawal of ipilimumab only | 1 (3.2) | |
| Any event leading to temporary discontinuation or withdrawal of both trial drugs | 15 (48.4) | |
| Any treatment-related event leading to withdrawal of both study drugs | 3 (9.7) | |
| Immune-related Hepatitis | 2 (6.5) | |
| Immune-related Colitis | 1 (3.2) | |
| Any event leading to patient death | 3 (9.7) | |
| Sepsis | 2 (6.5) | |
| Suicide | 1 (3.2) | |
| Any event occurring in ≥10% of patients | Any Grade | Grade 3 or 4 |
| Infection or sepsis | 10 (32.2) | 6 (22.5) |
| Nausea | 6 (19.4) | 1 (3.2) |
| Renal impairment | 6 (19.4) | 2 (6.4) |
| Anemia | 5 (16.1) | 0 |
| Vomiting | 5 (16.1) | 1 (3.2) |
| Constipation | 4 (12.9) | 1 (3.2) |
| Diarrhea | 4 (12.9) | 1 (3.2) |
| Fatigue | 4 (12.9) | 0 |
| Deterioration of general condition | 4 (12.9) | 3 (9.7) |
| Hypokalemia | 4 (12.9) | 3 (9.7) |
| Immune-related AEs | 14 (45.2) | 9 (29) |
| Hepatitis | 4 (12.9) | 4 (12.9) |
| Diarrhea / Colitis | 3 (9.7) | 3 (9.7) |
| Hyperthyroidism | 3 (9.7) | 0 |
| Hypophysitis | 1 (3.2) | 1 (3.2) |
| Arthritis | 1 (3.2) | 1 (3.2) |
| Pruritus | 1 (3.2) | 0 |
| Fatigue | 1 (3.2) | 0 |

Source data are provided as a Source Data file.

but eventually not included into the trial. The median baseline ccfDNA concentration in this cohort was 5.2 ng/ml plasma (interquartile range (IQR): 2.3–14.9 ng/ml plasma) and was significantly higher than in 19 healthy controls (1.7 ng/ml plasma, IQR 1.4–2.8 ng/ml, $p < 0.0003$; Fig. 4a). Within the CUP cohort, large differences between ccfDNA concentrations were observed (range 0.8–261.2 ng/ml plasma), with higher baseline ccfDNA concentrations (≥5.2 ng/ml) predicting for inferior survival (PFS, log-rank $p = 0.014$, HR 2.78, 95% CI 1.19–6.53; OS, log-rank $p = 0.008$, HR 3.05, 95% CI 1.29–7.22) (Fig. 4b, Supplementary Fig. 11a). ccfDNA levels were not associated with metastatic burden ($p = 0.157$, Supplementary Fig. 12a, b). From 13 CUP patients treated with nivolumab/ ipilimumab longitudinal follow-up samples were available for analysis (range, 1–9 per patient) (Fig. 1a). In total, we obtained 81 plasma samples for ccfDNA/ctDNA analysis and ICI response monitoring.

For ctDNA quantification, we designed a customized NGS panel based on the results from comprehensive genomic profiling of baseline FFPE tumor biopsies. This panel was tailored to target 61 patient-specific mutation hotspot regions in 41 different genes mutated in the study population (Supplementary Table 5), spanning only approximately 12.800 bp of the human genome, including complete exon coverage of *TP53*, and thereby allowing for ultra-deep targeted sequencing (VAF ≥ 0.1%). This strategy revealed a high correlation between mutation spectra of matched baseline plasma and tissue biopsy samples

($p < 0.0001$), with 65 of 77 tumor SNVs/indels detected in FFPE material being recovered in 29/34 (85.3%) study patients (Supplementary Fig. 11b). Comparison of baseline ccfDNA concentrations with ctDNA variant allele frequencies (VAF) revealed a statistically significant correlation ($p = 0.0243$; Supplementary Fig. 11c). In suspicious cases, whole exome sequencing of peripheral blood mononuclear cells was performed, confirming that in six patients some of the SNVs/indels detected in tumor tissue and ccfDNA were heterozygous germline mutations. Because only patients with at least one somatic mutation identified were considered for longitudinal ctDNA analysis, ultra-deep targeted NGS was able to detect ctDNA in 73.5% (25/34) of patients (Fig. 4c).

In five patients, longitudinal targeted NGS of ctDNA could not be performed because no somatic SNVs/indels were found in the baseline FFPE tumor material. To circumvent this problem, plasma samples were additionally analyzed for tumor-specific CNAs via sWGS of ccfDNA (Supplementary Figs. 13, 14, 15a). With this method, tumor fractions (TFx) > 4% could be identified in 21 baseline and 17 follow-up samples from 23/34 (67.7%) of patients (Fig. 4c). Importantly, sWGS allowed for the detection of CNAs in five patients without detectable somatic SNVs/indels. Vice versa, targeted NGS identified SNVs/indels in 28 samples from seven patients without tumor-specific CNAs (Fig. 4c, Supplementary Fig. 15b). Results of ctDNA measurements by sWGS and targeted NGS in matched samples were highly concordant ($n = 29$, $p < 0.0004$; Supplementary Fig. 15c). In total, our combined targeted NGS/ sWGS sequencing strategy was able to detect and quantify ctDNA in 88.2% (30/34) of CUP patients (Fig. 4c).

The median baseline ctDNA concentration was 74.6 haploid genome equivalents (hGE)/ml plasma (IQR: 5.0-281.8 hGE/ml plasma) and ranged from below the detection limit to 45,700 hGE/ml plasma. Although ctDNA and ccfDNA concentrations statistically significantly correlated with each other ($p = 0.0054$; Fig. 4d), high baseline ctDNA levels ( ≥ 37.4 hGE/ml plasma) were—in contrast to ccfDNA levels—not associated with significantly poorer survival of CUP patients (OS, $p = 0.834$; PFS, $p = 0.609$; Fig. 4e). Furthermore, baseline ctDNA levels did not correlate with either the sum of target lesions ($p = 0.984$) or the metastasis burden score ($p = 0.893$; Supplementary Fig. 12c, d). Additionally, we calculated the aneuploidy score from baseline ctDNA. In contrast to CNA profiling from FFPE tissue samples (Fig. 3c), no correlation between survival and aneuploidy was observed, likely due to low baseline ctDNA levels that prevented CNA profiling by sWGS in some patients (Supplementary Fig. 10b).

## ctDNA monitoring as a biomarker for response and survival in CUP patients receiving ICI therapy

By applying the above combined targeted NGS/sWGS sequencing strategy, we analyzed the dynamic changes in ctDNA levels from plasma samples serially collected in parallel with radiological assessment according to RECIST v1.1 during combined ICI treatment (Fig. 5a). In two of 13 CheCUP patients with follow-up samples, ctDNA was not detectable at baseline or disease progression three months after nivolumab/ipilimumab treatment by neither targeted NGS nor sWGS. Paired analysis of baseline and follow-up plasma samples from the remaining eleven patients demonstrated that our ctDNA monitoring strategy enabled reliable early-on discrimination already at first follow-up only three months after treatment initiation between patients who responded to ICI therapy ($n = 5$) and those suffering disease progression ($n = 6$) (Fig. 5b, c; Supplementary Fig. 16). Whereas patients with radiologic disease progression showed a marked increase in ctDNA levels compared to baseline samples, tumor shrinkage or stable disease was reflected by a decrease in ctDNA levels.

In all patients with detectable aberrations in both targeted NGS and sWGS, serial tumor-specific CNA profiling by sWGS paralleled the dynamics of somatic tumor SNVs detected by targeted NGS, underlining the usefulness of both approaches to monitor treatment outcome. For example, during the course of combined ICI therapy, both

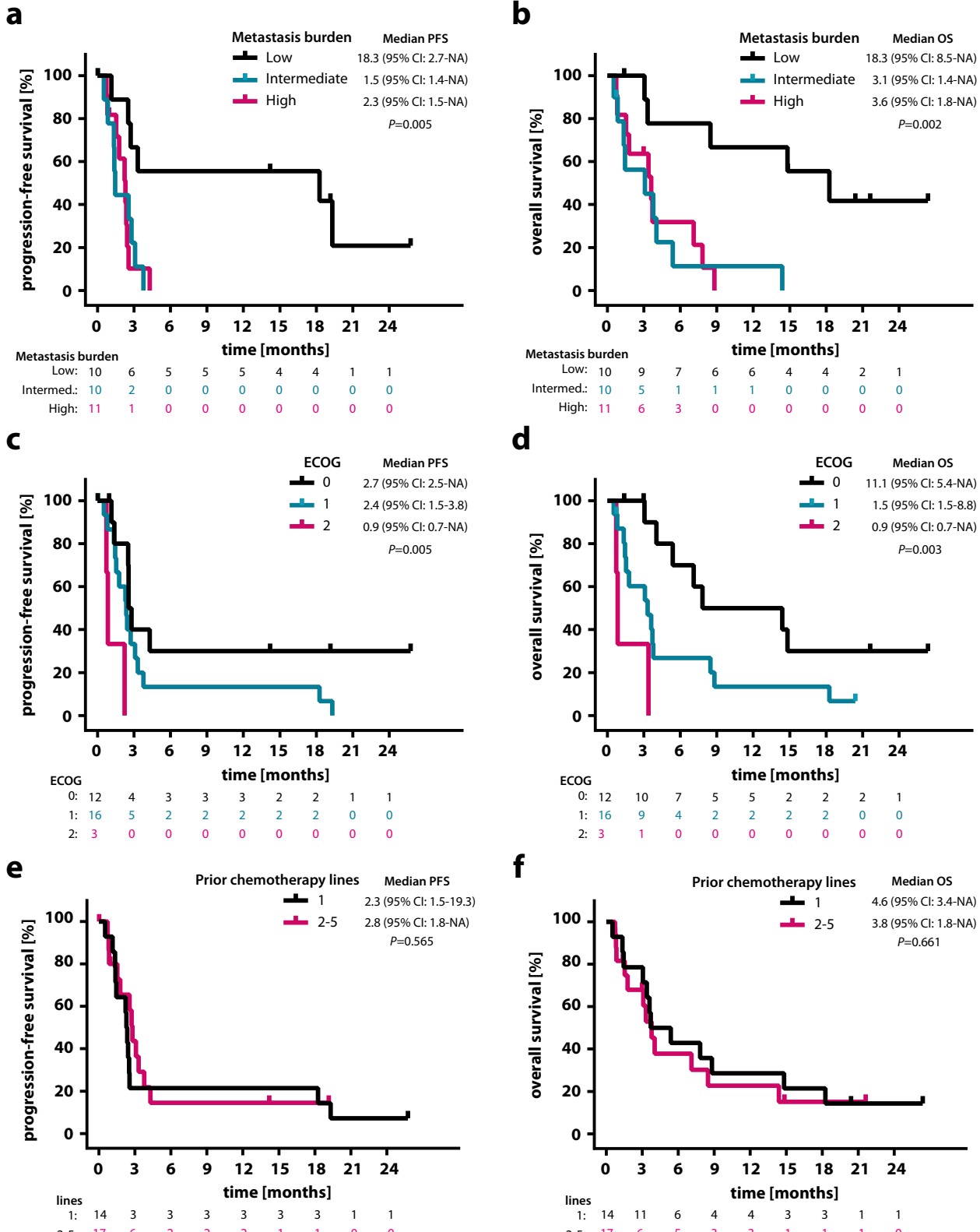

**Fig. 2 | High metastasis burden and high ECOG status were associated with increasing risk for progression and death.** Kaplan-Meier estimates of (**a**) PFS and (**b**) OS, classified according to low (*n* = 10), intermediate (*n* = 10) and high (*n* = 11) metastasis burden score. Kaplan-Meier estimates of (**c**) PFS and (**d**) OS, classified according to ECOG performance status: 0 (*n* = 12), 1 (*n* = 16) and 2 (*n* = 3). Kaplan-Meier estimates of (**e**) PFS and (**f**) OS, classified according to number of previous chemotherapy lines: 1 (*n* = 14) versus 2–5 (*n* = 17). Comparisons are made using a two-sided log-rank test, Cox proportional hazard regression modeling was used to calculate hazard ratio. 95% CI, 95% confidence interval; HR, hazard ratio; ECOG Eastern Cooperative Oncology Group. Source data are provided as a Source Data file.

**Table 4 | Multivariate analysis of progression-free and overall survival for the study population**

| | Progression-free survival | | Overall survival | |
|---|---|---|---|---|
| | HR (95% CI) | *P* value | HR (95% CI) | *P* value |
| Overall study population (*n* = 31) | | | | |
| TMB[high] | 0.70 (0.18–2.67) | 0.60 | 0.45 (0.11–1.79) | 0.26 |
| ECOG 1 | 2.00 (0.76–5.27) | 0.16 | 2.92 (1.07–7.98) | 0.04 |
| ECOG 2 | 5.92 (1.27–27.59) | 0.02 | 8.05 (1.60–40.50) | 0.01 |
| Intermediate disease burden | 4.43 (1.25–15.76) | 0.02 | 10.61 (2.50–45.04) | 0.0014 |
| High disease burden | 4.87 (1.28–18.60) | 0.02 | 5.30 (1.30–21.61) | 0.02 |
| Patients with available PD-L1 status (*n* = 15) | | | | |
| TMB[high] | 0.49 (0.10–2.46) | 0.39 | 0.31 (0.06–1.58) | 0.16 |
| PD-L1 positive | 0.71 (0.20–2.50) | 0.60 | 0.30 (0.07–1.22) | 0.09 |

The analysis comprised a multivariate Cox regression model that included the relevant prognostic factors. Source data are provided as a Source Data file.

CNAs and the TERT promoter C250T mutation present at baseline in ctDNA of patient P4 were no longer detected in follow-up samples, whereas in patient P6, both CNAs and one tumor-specific mutation (KMT2D-p.Q2337*) undetectable at baseline became recognizable at disease progression (KMT2D VAF: 1%; TFx estimate: 4.6%), and increased massively before the patient's death (KMT2D VAF: 9.8%; TFx, estimate: 12.7%; Supplementary Fig. 17).

We conclude that longitudinal ctDNA monitoring by combined targeted NGS and CNA profiling can assist in early identification of patients deriving long-term ICI treatment benefit, and is particularly useful in patients with initial radiologic stable disease. In such cases, complete response to ICI therapy, which became visible by radiologic imaging only much later, could be predicted already at first response assessment using ctDNA monitoring. Accordingly, in two cases (patient P1 and P16), we observed a complete clearance of ctDNA already three months after treatment initiation. Radiologically, patient P1 (Fig. 5d) achieved a partial and complete response at 12 and 24 months, respectively, and was still in complete remission at the time of database lock after 27 months of combined nivolumab/ipilimumab administration, whereas patient P16 (Supplementary Fig. 18) reached radiological complete remission at 18 months. Vice versa, although patient P12 reached radiological complete remission at six months of ICI administration, targeted NGS still detected a somatic ICI resistance-associated *PIK3CA* p.E545K mutation[49] in ccfDNA, which remained stable during continued ICI therapy (Fig. 6a, b). Although until data cut-off no radiological disease progression was reported, this patient remains under close ctDNA and radiological surveillance. Tumor-specific CNA profiling of ccfDNA from patient P19 showed that despite decline, CNAs remained detectable at first follow-up (TFx estimate: 6.7%) and gradually increased in subsequent samples (up to TFx estimate of 15.8%), allowing prediction of disease progression from partial remission six months prior to radiological relapse (Fig. 6c, d).

## Discussion

This investigator-initiated phase II trial assessed combined immunotherapy with nivolumab and ipilimumab in patients with unfavorable CUP relapsed after or refractory to platinum-based chemotherapy. Combined immunotherapy led to an ORR of 16.2% in the total study population of 31 patients, with 60% of TMB[high] patients but only 7.7% of TMB[low] patients responding. Median PFS, the primary endpoint of the trial, was 18.3 versus 2.4 months for TMB[high] and TMB[low] patients, respectively. Main scope of the trial was to investigate the role of TMB and ctDNA as predictive biomarkers for response and survival to immune checkpoint inhibition.

The ORR of 16.2% found in our study is comparable but not superior to previously published data on nivolumab (ORR 22.2%) and pembrolizumab (ORR 20%) monotherapy in two other comparatively

small cohorts of relapsed/refractory CUP patients[50,51]. Although one of these trials[51] led to the approval of nivolumab for both untreated and pretreated CUP patients by the Japanese Pharmaceuticals and Medical Devices Agency (PMDA), the study was criticized because of concerns on reproducibility due to its small sample size and the lack of prospective predictive biomarker analysis[52]. Our data confirm these earlier immunotherapy results in pretreated CUP patients. In addition, prospective stratification into TMB[high] versus TMB[low] patient cohorts in our trial now suggests that the benefit from checkpoint inhibitor therapy in pretreated patients with unfavorable CUP seems to be mainly restricted to the TMB[high] cohort.

The predictive value of TMB in the context of immunotherapy has been previously demonstrated in several entities[17]. Accordingly, in CUP patients with high TMB, defined as at least 12 mutations/Mb, the combination of nivolumab and ipilimumab led to an ORR of 60% and induced a durable benefit, as 60% of these patients were progression-free at one year versus ORR and one-year-PFS of 7.7% and 9%, respectively, in the TMB[low] group. In addition to the clear trend for an increased median PFS (18.3 vs. 2.4 months), immunotherapy led to an increased median OS in TMB[high] compared to TMB[low] patients (18.3 vs. 3.6 months), again without quite reaching statistical significance, likely due to the limited sample size.

In our total cohort of 31 platinum-pretreated patients, median PFS and OS were 2.53 and 3.8 months, respectively, which is lower than previously published data reporting a median PFS of 4–4.1 and OS of 11.3–15.9 months after second-line immunotherapy[50,51]. Of note, 45% of patients in our cohort suffered from high disease burden, explaining the particularly aggressive course of their disease. As a result, 42% of patients discontinued treatment during the first six weeks, mostly due to deterioration of their clinical condition because of rapid disease progression, and consequently received no more than one treatment cycle. Therefore, and in line with similar observations in other tumor entities[53–55], although immunotherapy is effective in CUP, patients with high disease burden and poor performance status due to rapid progression appear, analogous to other cancer types, to benefit to a lesser extent from immune checkpoint inhibition[56–60]. On the other hand, in patients with MSI[high] cancers immunotherapy has been reported to be effective regardless of performance status[61]. Although a favorable effect of TMB[high], particularly on OS, was sustained to some extent in multivariate analysis of our study population, ECOG and disease burden slightly weakened the effect of TMB on PFS. Both lower ECOG and lower disease burden were more frequent in the TMB[high] group, which might account for the reduced effect of TMB in multivariate analysis. However, due to the limited sample size and the resulting low *events per variable* ratio, the analysis remains inconclusive regarding the predictive value of TMB in CUP patients with poor performance status and high disease burden[62,63].

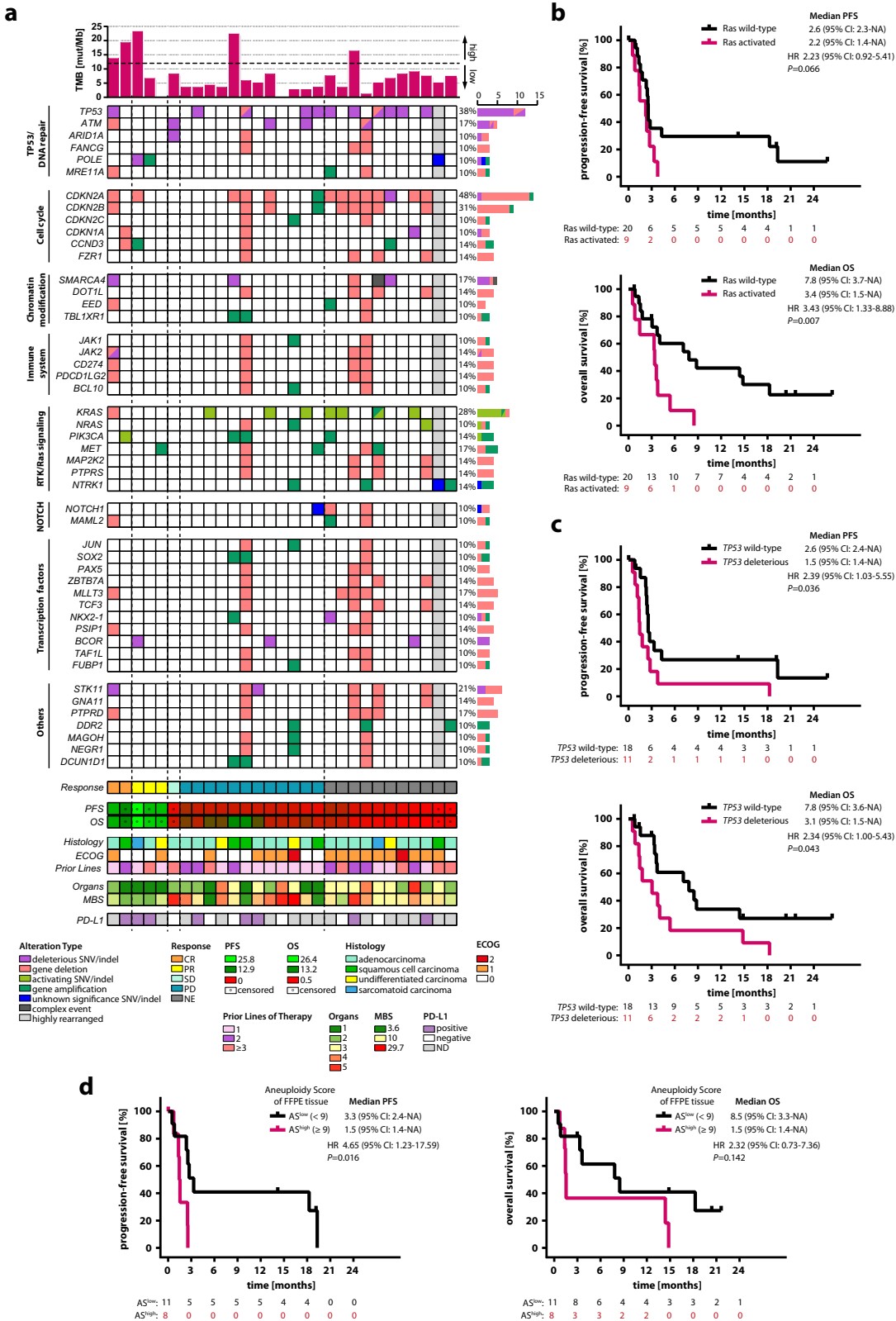

In addition to TMB, retrospective data suggest that high tumor aneuploidy levels are associated with poor outcome following ICI treatment across multiple cancer types with low but not high TMB[47]. Of note, a higher aneuploidy score was associated with a significantly shorter PFS but not OS both in the total and the TMB[low] CheCUP population, when aneuploidy was calculated from tumor tissue. On the other hand, no association between survival and aneuploidy was found when the aneuploidy score was calculated from ctDNA, probably because in some patients low baseline ctDNA levels prevented CNA profiling by sWGS.

While PD-L1 expression was associated with a trend towards prolonged OS as well, the TMB effect persisted after adjustment for PD-L1, which suggests that its predictive value might be independent of PD-L1 expression. Similar constellations have been previously

**Fig. 3 | Baseline genomic landscape of the CheCUP cohort revealed that patients with activated Ras signaling and/or functionally deleterious *TP53* did not benefit from ICI treatment. a** Oncoplot showing potentially clinically relevant tumor gene alterations (SNVs/indels, gene deletions and amplifications) as assessed by comprehensive genomic profiling of baseline biopsy samples. Curated pathways and selected genes altered in 10% or more patients are shown. A column represents a patient and is grouped by the best response (indicated by vertical dashed lines) and related clinicopathologic features (n = 29); gray shaped columns indicate patients where an unambiguous CNA profile statement was impossible. Top bar chart represents tumor mutational burden (TMB). Percentages listed right represent the proportion of patients harboring an alteration in the gene listed left. Bottom bars show radiological response assessment, heatmaps of PFS and OS (both in months, censored patients marked with dots), CUP histology, ECOG status,

number of therapy lines prior to ICI treatment, number of organs with metastases, heatmap of metastasis burden score (MBS), and PD-L1 expression status. **b** Kaplan-Meier estimates of PFS and OS, stratified according to activating *RAS* alterations: activated (n = 9) and wild-type (n = 20). **c** Kaplan-Meier estimates of PFS and OS, stratified according to functionally deleterious *TP53* alterations: deleterious (n = 11) and wild-type (n = 18). **d** Kaplan-Meier estimates of PFS and OS, stratified according to low (n = 11) or high (n = 8) aneuploidy score (AS) of baseline FFPE tumor tissue, based on the median cohort AS value of 9. In (**b-d**), comparisons are made using a two-sided log-rank test, Cox proportional hazard regression modeling was used to calculate hazard ratio. 95% CI 95% confidence interval, HR hazard ratio, CR complete response, PR partial response, SD stable disease, PD progressive disease, FFPE formalin-fixed paraffin-embedded, AS aneuploidy score, NE not evaluable, ND not determined. Source data are provided as a Source Data file.

described in several other entities including non-small cell lung cancer, where dual checkpoint inhibition with nivolumab and ipilimumab significantly prolonged PFS and OS particularly in patients with high TMB regardless of PD-L1 expression[17,64,65]. In unfavorable CUP, previous studies have reported a correlation between PD-L1 expression levels and ORR[50,51] as well as PFS and OS[51]. However, in these signal-finding studies biomarker analysis was performed retrospectively only and none of these trials conducted combined biomarker analysis in order to discern the individual effects of these factors. Despite the small sample size, our trial provides prospective evidence of TMB as an independent biomarker of immunotherapy efficacy in CUP. Importantly, MSI, which has been incriminated to compromise the validity of a nivolumab monotherapy trial in CUP[52] as it is known to predict treatment response irrespective of the cancer tissue of origin[66], was not detected in any of our patients and can therefore be excluded as a confounding factor in the current trial. That being the case, our data underline the role of high TMB as a predictive marker for immunotherapy in CUP.

The immune cell composition of the tumor microenvironment is known to impact on response to ICI therapy[67]. In a subgroup of CheCUP patients for whom sufficient material for gene expression-based quantification of tumor-infiltrating immune cell populations was available, none of the cell types detected was associated with patient survival. However, a significant correlation with TMB was observed for neutrophils and Treg cells, the presence of which has generally been associated with reduced survival and poor response to ICI treatment. As outside of the ICI treatment setting higher TMB levels are associated with poorer survival in many cancer types[68], recruitment of neutrophils and Treg cells into TMB^high tumors might contribute to this phenomenon.

Two randomized clinical trials suggest that gene expression profiling-based tissue-of-origin prediction with subsequent primary site-directed treatment is not superior to platinum-based chemotherapy in the first-line treatment of patients with unfavorable CUP[3,8]. Similarly, putative primary sites did not impact on response and survival of the CheCUP patient population following ICI treatment, when tissue-of-origin was determined based on clinical, radiological and immunohistochemical features.

Additional important questions remain regarding the role of combination strategies and optimal sequence of therapies. With regard to combination treatment, dual immune checkpoint blockade with nivolumab and ipilimumab failed to improve efficacy as compared to nivolumab or pembrolizumab monotherapy[50,51], which is in line with the results of a recent metaanalysis that showed no additional survival benefit through the addition of ipilimumab to standard-dose nivolumab in advanced cancers other than melanoma[69]. However, since the patient cohorts treated within both CUP monotherapy trials slightly differed from the CheCUP cohort in terms of ECOG status and previous treatment lines, randomized data are needed to conclusively solve this point. Data for immune-chemotherapy combinations, which

have been shown to be active in patients not expected to benefit from single-agent immunotherapy in other tumor entities[70-72], are not yet available in CUP but are currently being tested in the CUPISCO trial[73]. Regarding treatment sequence, the benefit of immunotherapy was independent of the number of previous therapy lines, making it a valuable option even for heavily pretreated patients.

The limited sample size constitutes a main limitation of the CheCUP trial. Due to slower than expected recruitment, the trial was terminated early by the sponsor, thereby highlighting once more and similar to earlier attempts the accrual challenge with CUP trials even in a multicenter setting. As a result, the trial was underpowered and all analyses were exploratory. Nevertheless, the study does offer meaningful insights into the value of immunotherapy in CUP as, despite these limitations, both hazard ratio and PFS approximated the initially assumed values.

For early response monitoring, we used combined quantitative ultra-deep targeted NGS of patient-specific hotspot tumor mutations and tumor-specific CNA profiling by low-coverage sWGS of ccfDNA in comparison to radiologic response evaluation. The targeted NGS part of our approach is able to detect very low amounts of ctDNA but relies on a specifically designed fingerprint panel that only covers patient-specific hotspot mutations previously identified by mutational profiling of tumor tissue. As a limitation, this technique does neither capture metastatic heterogeneity nor newly acquired mutations during immunotherapy.

Combined targeted NGS/sWGS allowed for longitudinal ctDNA monitoring in 90% of CUP patients. ctDNA dynamics paralleled radiological response assessment, indicating that ctDNA evaluation might be a valuable surrogate of treatment response in CUP. As the timing of ctDNA sample collection was predefined to occur in parallel to radiological response assessment in three-monthly intervals, trial data are not suited to determine the optimum timepoints of ctDNA analysis for prediction of immunotherapy response. Nevertheless, on-treatment ctDNA kinetics three months after immunotherapy initiation identified patients achieving long-term remission when radiologic findings still indicated stable disease. Since radiologic stable disease can stretch from minimal response to beginning progression, ctDNA monitoring might enable early-on discrimination between patients who will eventually derive treatment benefit and those with progressive disease. Given the toxicity profile of immunotherapy, the development of biomarkers predicting treatment response is essential not least to spare toxicities.

In conclusion, combined ipilimumab/nivolumab treatment led to high response rates in patients with unfavorable CUP relapsed or refractory to platinum-based chemotherapy and a TMB of at least 12 mutations/Mb, irrespective of PD-L1 expression levels and number of previous chemotherapy lines. Early reduction of ctDNA levels from baseline predicted benefit from immunotherapy better than radiologic tumor burden assessment. These results underline the role of immunotherapy as an effective second-line treatment in CUP, and TMB as

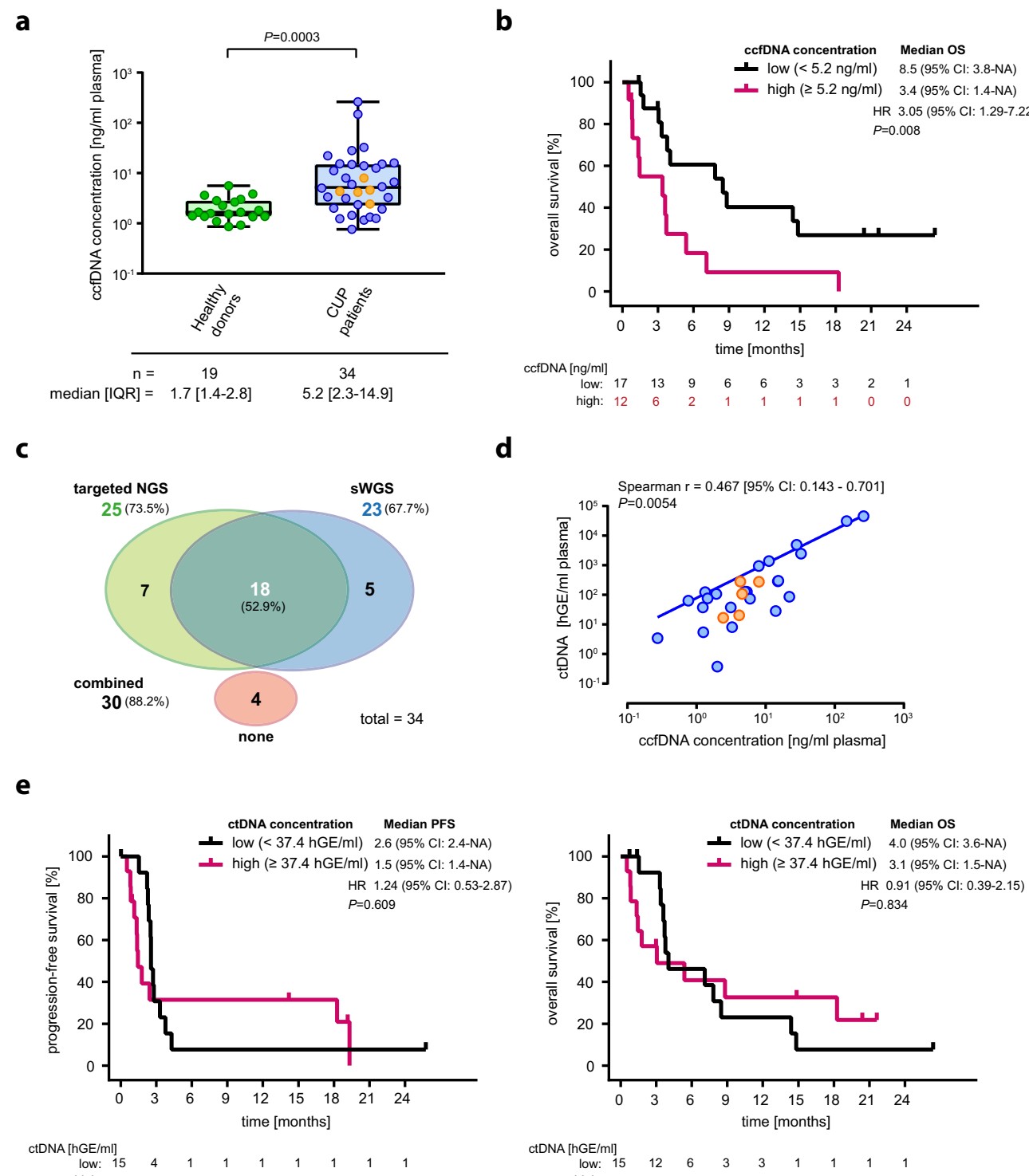

well as ctDNA kinetics as independent predictors of benefit from ICI treatment.

## Methods
### Patient cohort
All patients provided written informed consent to the trial and to the accompanying translational research (NCT04131621; EudraCT No. 2018-004562-33). The trial was preregistered on the EU Clinical Trials Register on 22.07.2019 and on the ClinicalTrials.gov registry of the United States National Library of Medicine on 18.10.2019. The pre-registered protocol version (V1.7) can be accessed under https://www.

clinicaltrialsregister.eu/ctr-search/trial/2018-004562-33/DE. Protocol amendments included clarifications and changes to inclusion and exclusion criteria to enhance subject enrollment e.g. extension of screening period, acceptance of pre-existing TMB analysis (not older than six months) or pre-existing tumor FFPE tissue blocks, if a re-biopsy failed for inclusion, inclusions of subjects with limited brain metastases and COVID-19 triggered changes, e.g. allowance of local blood sample analysis and possible remote patient visits, if no IMP is administered. The protocol was approved by the local ethics com-mittees of the participating centers, the leading ethics committee of the Medical Faculty of the University of Heidelberg (date of approval:

**Fig. 4 | Elevated baseline ccfDNA but not ctDNA is prognostic for inferior overall survival and ICI response in patients with recurrent/refractory unfavorable CUP. a** Baseline ccfDNA concentrations of healthy individuals ($n = 19$) and patients with recurrent or relapsed unfavorable CUP ($n = 34$). Each dot represents a single healthy individual or CUP patient, respectively; orange dots indicate CUP patients who radiologically respond to ICI treatment. The box contains the 25th to 75th percentiles of the dataset, a vertical line denotes the median value. The whiskers go from first or third quartile to the minimum or maximum values, respectively. An exact $p$-value ($p = 0.0003$) was calculated using a two-tailed Mann–Whitney U test. Of note, the y-axis is in log-scale for display purpose only. **b** Kaplan-Meier estimate of OS, stratified according to high ($n = 12$) or low ($n = 17$) ccfDNA based on a ccfDNA concentration cut-off of 5.2 ng/ml plasma. **c** The efficiency of the combined targeted/sWGS sequencing strategy in detecting ctDNA in blood samples of CUP patients. **d** Scatter plot of baseline ccfDNA concentration versus baseline ctDNA content of unfavorable CUP patients ($n = 34$). Each dot represents a single patient; orange dots indicate CUP patients who radiologically respond to ICI treatment. Correlation analyses were performed based on Spearman's correlation coefficient; a regression line was fitted. Of note, the log-log axes are for display purpose only. **e** Kaplan-Meier estimates of PFS and OS, according to high ($n = 14$) or low ($n = 15$) ctDNA based on a ctDNA level cut-off of 37.4 hGE/ml plasma. In (**b, e**), crosses denote censored observations, and for each time interval the number of patients at risk are indicated below the plots. Comparisons are made using a two-sided log-rank test, Cox proportional hazard regression modeling was used to calculate hazard ratio. 95% CI 95% confidence interval, HR hazard ratio, IQR interquartile range, hGE haploid genome equivalent. Source data are provided as a Source Data file.

27.11.2019), and the competent authorities (date of approval by the Paul Ehrlich Institute: 04.12.2019). The final study protocol (V1.9) is provided as a Supplementary Note in the Supplementary Information file. The trial was conducted in accordance with the Good Clinical Practice guidelines and the Declaration of Helsinki. The study design and conduct complied with all relevant regulations regarding the use of human study participants. An independent data and safety monitoring committee provided oversight of safety and efficacy. Funding and investigational drugs were provided by Bristol-Myers Squibb (BMS study ID: CA209-8WY).

Adult patients with histologically confirmed disseminated or advanced unresectable CUP diagnosed according to the ESMO clinical practice guidelines[1], who had experienced disease progression or relapse after at least three cycles of platinum-based chemotherapy were eligible. Acceptable histology included adenocarcinoma, poorly differentiated adeno-carcinoma, poorly differentiated carcinoma and squamous cell carcinoma. In order to determine TMB status, either fresh or archival (≤ 6 months) tumor FFPE samples sufficient for the generation of a TSO500 panel had to be provided. Eligible patients were required to meet the following criteria: at least one lesion that is measurable according to RECIST v1.1 and has not been irradiated, ECOG performance status 0–2, adequate hematologic and end-organ function (ANC $\geq 1.0 \times 10^9$/l, platelet count $\geq 80 \times 10^9$/l, hemoglobin $\geq$ 90 g/l, AST and ALT $\leq$ 3x upper limit of normal (ULN), serum bilirubin $\leq$ 1.5xULN, creatinine clearance $\geq$ 30 ml/min, INR and aPTT $\leq$ 1.5 x ULN for patients not receiving therapeutic anticoagulation). All patients underwent imaging to screen for brain metastases. Patients with brain metastases were initially excluded. The protocol was amended in April 2020 to include patients with no more than three asymptomatic CNS metastases, provided the metastases had been completely surgically or radiosurgically treated with no evidence of residual disease at screening. Patients had to be eligible to immune checkpoint inhibition and demonstrate no concomitant diseases or autoimmune diseases, physical examination findings or laboratory findings that contraindicate the use of immunotherapy or require systemic treatment with either corticosteroids or other immunosuppressive agents. Patients belonging to any of the favorable CUP subsets according to the ESMO guidelines were excluded. The first and last patients were enrolled on December 12th 2019 and February 22nd 2021, respectively.

**Trial design, endpoints and assessments, sample size calculation**
The CheCUP trial was investigator-initiated and was designed as a phase II, open-label, non-randomized, multi-center study evaluating the efficacy and safety of combined nivolumab and ipilimumab in previously treated patients with CUP. The trial was planned to include 194 patients and was performed at ten sites throughout Germany. After failure of platinum-based therapy, eligible patients were stratified based on TMB (high vs low, cut-off 12 mutations/Mb) following comprehensive genomic profiling and treated with combined nivolumab (240 mg biweekly) and ipilimumab (1 mg/kg every 6 weeks) until

disease progression, loss of clinical benefit, unacceptable toxicity, death from any cause or withdrawal of consent. Treatment continuation beyond progression was permitted if the patient had investigator-assessed clinical benefit and continued to tolerate treatment. Patients were followed up for overall survival and safety objectives at day 100 and every 3 months thereafter via phone contact after discontinuation of study drug treatment. The study design aimed at balanced strata sizes with 50% of patients belonging to the TMB[high] and 50% belonging to the TMB[low] group.

The primary endpoint was progression-free survival (PFS), defined as the time from treatment start to the first occurrence of disease progression as assessed by the investigator according to RECIST v1.1 or death from any cause, whichever occurs first. Secondary endpoints included overall survival, defined as the time between start of therapy and death from any cause, overall response rate (ORR), defined as the proportion of patients who exhibit a complete (CR) or partial response (PR) to study treatment and duration of clinical benefit (DOB), defined as the time from the first occurrence of a CR, PR or stable disease (SD) after treatment start until disease progression or death from any cause. Safety and tolerability endpoints included incidence, nature and severity of adverse events (AEs), incidence and reasons for any dose reductions, interruptions or premature discontinuation of any component of the study treatment, as well as clinically significant laboratory values and vital sings.

Sample size was calculated with R package gsDesign (https://CRAN.R-project.org/ package=gsDesign) according to Lachin and Foulkes (1986)[74]. A total of 194 CUP patients with 191 events were required to detect a hazard ratio of 0.65 for TMB[high] vs TMB[low] patients with 80% power at the two-sided significance level of 5%. Due to previous studies[11,12,15,19,20], median PFS was assumed to be 2.3 months and 15% of subjects were expected to have a high TMB status. TMB[high] patients were expected to have a favorable prognosis. Assuming a hazard ratio of 0.65 and exponentially distributed survival, median survival times were assumed to be 2.18 and 3.35 months for TMB[low] and TMB[high] subjects, respectively. Since patients were considered to be recruited in a 1:1 ratio, the TMB[high] patients had to be enriched by screening approximately 700 patients. A 24-months recruitment period and a minimal follow-up time of 12 months were planned to allow observing the necessary number of events under the above stated assumptions. Due to insufficient enrollment of patients in the first year, financial support and supply of study medication were withdrawn and the trial had to be terminated early in March 2021. As a result, TMB groups were ultimately unbalanced.

The study was conducted on a national, multicenter basis and took place at the following German sites: University Hospital Jena, Evangelische Kliniken Essen-Mitte, Marienhospital Stuttgart, Kliniken Ostholstein ohO Onkologie, Uniklinikum Augsburg, Klinikum der Universität München, Universitätsklinikum Leipzig, Universitätsklinikum Tübingen, Onkologische Praxis Gütersloh, Universitätsklinikum Heidelberg. All sites are large university hospitals, community hospitals or oncological practices with dedicated, high-

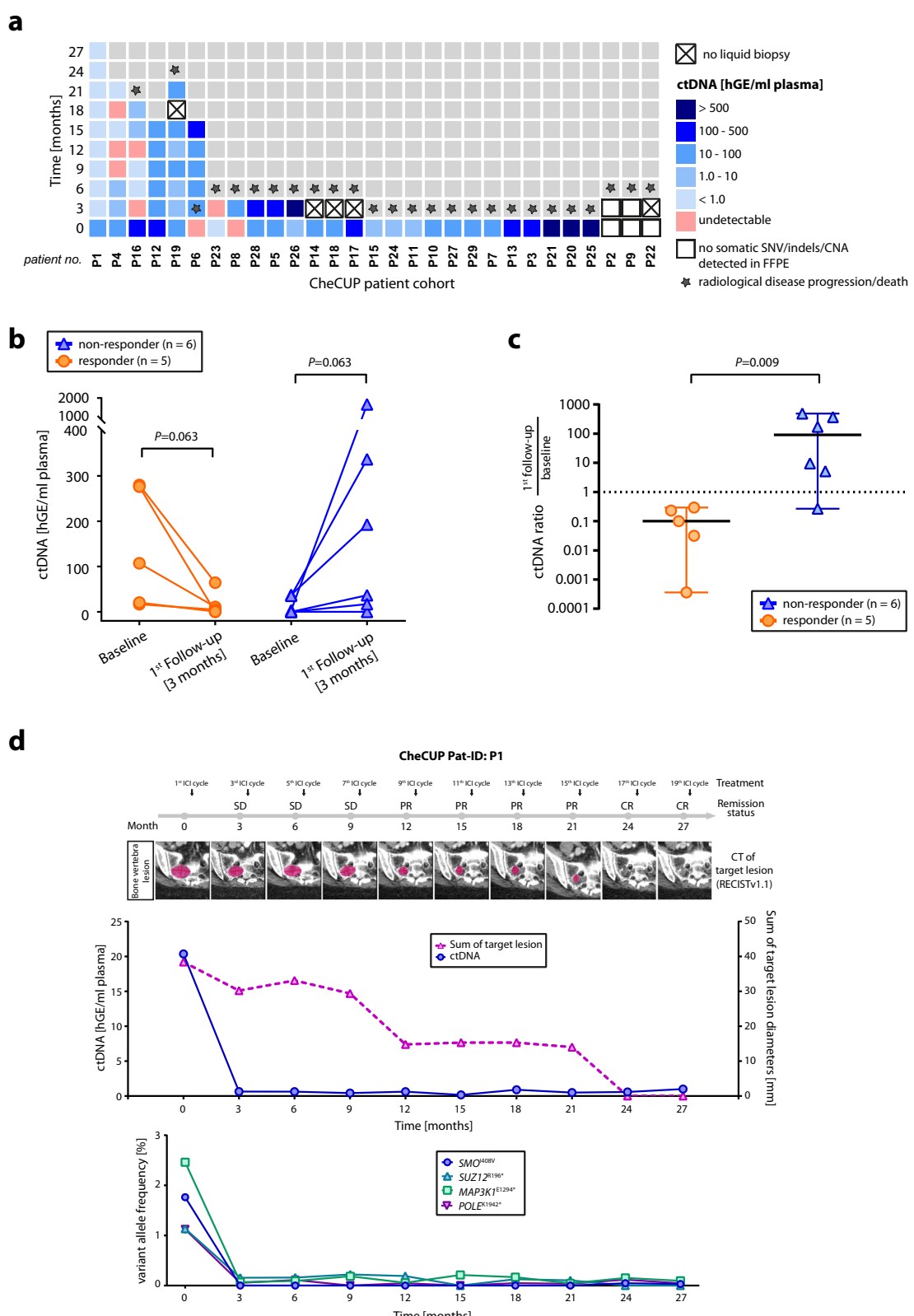

volume oncology units and large numbers of CUP subjects. In addition, each of the sites has a clinical trial center and study nurses specifically dedicated to execution and documentation of the CheCUP study.

For the purpose of this study, sex as a biological attribute was determined based on self-reporting. As there were no preferences on the selection of gender to be included, it was anticipated that the study will result in a representative gender distribution, which should reflect the natural gender distribution in the underlying disease. There are no study findings that apply to only one sex. Since no specific findings were expected according to gender, this information was not collected. Based on previous observations, sex was not expected to affect survival of the trial participants. For this reason, no analyses apart from descriptive frequency analysis were performed regarding sex.

**Fig. 5 | The benefit of longitudinal ctDNA monitoring parallel to radiological assessment in patients with recurrent/refractory unfavorable CUP. a** Time-measured ctDNA content of each CheCUP patient determined by combined ultra-deep targeted NGS of patient-specific hotspot mutations and sWGS-based CNA profiling in serially collected plasma samples during ICI treatment. **b** Change in ctDNA level in paired baseline and first follow-up plasma samples after three months of ICI treatment of responding (orange, *n* = 5) and non-responding (blue, *n* = 6) patients, respectively. Each dot/triangle represents a single CUP patient. Comparisons between baseline and first follow-up samples of the same patient were made using a two-tailed Wilcoxon matched-pairs signed rank test. All *p*-values are exact. **c** Comparison between the molecular response of radiologically responding (orange, *n* = 5) and non-responding patients (blue, *n* = 6). The molecular response was calculated as ratio of first follow-up to baseline ctDNA content.

Each dot/triangle represents the molecular response of a single patient; bold horizontal bars indicate the median values ±95% confidence interval; a dashed horizontal line indicates the cut-off predicting response and outcome. The comparison between the median of radiologically responding and non-responding was made using a two-tailed Mann–Whitney U test. Of note, the y-axis is in log-scale for display purpose only. **d** Representative case (CheCUP patient P1) exemplifying the utility of ctDNA analyses in monitoring ICI response in parallel to radiological assessment. Upper graph: Comparison of ctDNA level in serial collected plasma sample with the measured sum of target lesion diameters by RECIST v1.1. during combined nivolumab/ipilimumab administration. Lower graph: Dynamic tracking of VAF from single somatic tumor mutations in ctDNA. hGE haploid genome equivalent. Source data are provided as a Source Data file.

## Data collection

A clinical data management system provided by the Coordination Center for Clinical Trials (KKS) of the University of Heidelberg was used for data collection using an electronic CRF (eCRF) for remote data entry (RDE). All entries in the eCRF had to be verifiable by source documents. A detailed list of entries to the eCRF was provided in the Investigator Site File. Regardless, there had to be a minimum documentation, which provided information on study participation and included all medical information necessary for appropriate medical care outside of the clinical trial in the patient record. In addition, source documents had to mention that the subject had been included in an investigational study. Finally, there had to be no data that were inconsistent between eCRF and source documents. All protocol-required information collected during the trial had to be entered by the investigator or a designated representative into the eCRF. Patient data was documented pseudonymously. The investigator or a designated representative should complete the eCRF pages as soon as possible after the information was collected, preferably within two weeks after the study visit. Any pending entries had to be completed immediately after the final examination. Explanation had to be given for all missing data. The investigator was responsible for ensuring that all sections of the eCRF were completed correctly. Any errors had to be corrected in the eCRF and a reason for change had to be entered. The correctness of all entries in the eCRF had to be confirmed by dated electronic signature of the responsible investigator. The time points and frequency of electronic signatures were defined in the study-specific document "eCRF specification".

The primary data collection took place locally at the treating site. Data was collected from the data of informed consent until the discontinuation of study treatment. Following discontinuation of treatment, subjects were followed up 30 days after the last treatment or at initiation of another anti-cancer therapy (End of treatment visit). Patients were then contacted by their physician at day 100 for a safety follow-up and data regarding adverse events were recorded until day 100 after the final dose of study treatment. Thereafter, data regarding survival were collected every three months. Data was collected between December 12th 2019 (first patient in) and March15th 2022. After the final data check, the database lock took place on June 15th 2022.

## Clinical specimens

The diagnostic FFPE tumor tissue samples were used to perform a comprehensive baseline genomic profiling of metastatic lesion at the Center of Molecular Pathology (MPZ), Institute of Pathology, University of Heidelberg, Germany. Using the large-scale TruSight Oncology 500 (TSO500, Illumina) panel that covers 521 cancer relevant genes and comprises a total of 1.94 megabases (Mb) of the genome, the genomic profiles included not only potentially clinically relevant tumor mutations but also a statement about tumor mutational burden (TMB) and microsatellite instability (MSI). DNA extraction and libraries for the capture-based TSO500 were prepared as described previously[75,76]. The sequencing of up to eight TSO500

libraries was performed on a NextSeq500 platform (Illumina) using high-output cartridge and v2/v2.5 chemistry. Data analysis including TMB and MSI determination (based on a MSI cut-off of 10%) was carried out on a local docker version of the TSO500 Local App (Illumina, pipeline 1.3.0.39) as described previously[75]. In cases, where the metastatic tissue biopsy material was not enough to generate a TSO500 comprehensive genomic profile we have used the Oncomine Comprehensive v3 DNA panel (Thermo Fisher Scientific) that detects relevant SNVs, CNAs, gene fusions, and indels from 161 unique cancer-associated genes. DNA extraction, library preparation and semiconductor sequencing on the IonTorrent S5XL/Prime sequencing system for the Oncomine panel were conducted as described previously[77]. Data analysis was performed using the Ion Torrent Suite Software (versions 5.0.2.1 up to 5.2.2), variant calling using the variant caller plugin (version 5.0.2 to 5.2.20-1) and IonReporter package, and CNA screening using the Bioconductor package CNVPanelizer, as described previously[77].

PD-L1 expression status had been determined as part of the diagnostic workup prior to the inclusion to the CheCUP trial in ten cases. Additional five patients had FFPE tumor tissue available for PD-L1 immunohistochemistry, which was subsequently performed at the Institute of Pathology Heidelberg according to local standards. In brief, 3 µm thick paraffin sections were prepared. Deparaffinization and tissue staining were performed using a Ventana Benchmark Ultra device (Roche Diagnostics, Mannheim, Germany). Slides were deparaffinized and incubated with cell conditioning solution (Cell Conditioning 1 [CC1], Roche Diagnostics) at 95 C for 64 min. IHC staining was performed according to standard protocols (Ventana PD-L1 assay, clone SP263; Roche, Mannheim, Germany; Material number: 07208162001; Part Number: 740-4907; incubation time: 16 minutes). Hematoxylin was used for counterstaining of cell nuclei. IHC stainings were evaluated by a specialist in pathology and scoring of PD-L1 was performed according to standardized scoring criteria. PD-L1 positivity was defined as either Combined positivity score (CPS) or Tumor proportion score (TPS) ≥1.

Plasma samples were longitudinally collected from 34 CUP patients (median: 1; range: 1-10 samples per patient) starting at screening period prior ICI treatment and afterwards on day 1 of every second treatment cycle (approximately every three months after treatment initiation). In addition, 19 plasma samples from healthy donors were collected at the German Cancer Research Center of Heidelberg, Germany. All healthy donors provided written informed consent for the use of biological material for all study-related purposes by consenting to the preapproved Molecularly Aided Stratification for Tumor Eradication Research (NCT-MASTER) program (ClinicalTrials.gov ID NCT05852522). Plasma was isolated from whole blood two or 24 hours after venipuncture using the recommended double spin protocol for maximum plasma recovery. Briefly, 20 ml whole blood collected and stored in STRECK BCT tubes (Streck) was first centrifuged at 1600 x g for 10 min at room temperature. The upper plasma layer was again centrifuged at 4600 x g for 10 min at room

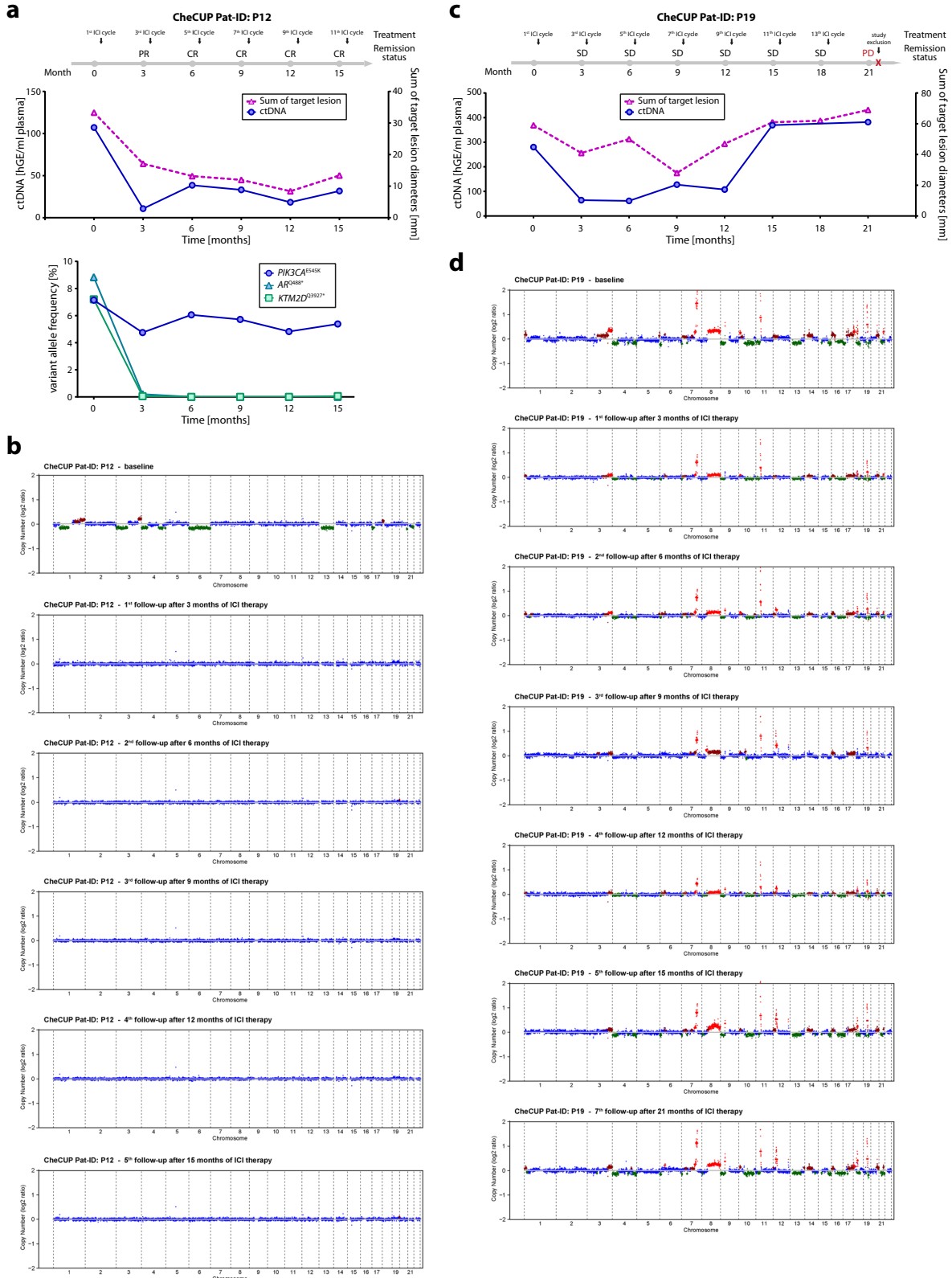

temperature to remove cell debris, and directly utilized for ccfDNA isolation as detailed below.

## Radiological response evaluation and metastasis burden calculation

Radiological response to combined nivolumab/ipilimumab was assessed in three months intervals after treatment initiation according to RECIST version 1.1 by reference radiologists in the field of CUP. Metastasis burden was defined as a function of the sum of diameters of all target lesions added to the score that resulted according to Supplementary Table 1 and depicted both the number of affected organs and the number of metastases per organ. According to this score, patients were then categorized into three groups (low, intermediate and high disease burden) with tertiles defined as cut-offs.

**Fig. 6 | Both targeted NGS of patient-specific hotspot mutations and tumor-specific CNA profiling by sWGS predicted ICI resistance/disease progression several months prior to radiological relapse. a, b** Monitoring of tumor burden and ctDNA changes in a case (CheCUP patient P12) with a stable detected *PIK3CA* p.E545K ICI-associated resistance mutation by targeted NGS. **a** Upper graph: Comparison of ctDNA level in serial collected plasma samples with the measured sum of target lesion diameters by RECIST v1.1. Despite radiological complete response, on-treatment ctDNA analyses by targeted NGS detected subliminal amount of ctDNA. hGE, haploid genome equivalent. Lower graph: Detection of a subclonal PIK3CA-p.E545K ICI-associated resistance mutation by dynamic tracking of VAF from single somatic tumor mutations. **b** Genome-wide CNA profiles inferred from sWGS were consistent with radiological response assessment. Chromosome

regions in shades of red indicate CNA gains, regions in green CNA losses. **c, d** Monitoring of tumor burden and ctDNA changes in a case (CheCUP patient P19) with an unique heterozygous germline mutation *CHEK2* p.T476M. **c** Comparison of ctDNA level, measured by sWGS-based CNA profiling, with the sum of target lesion diameters, measured by RECIST v1.1. ctDNA changes preceded clinical progression, while the disease was still radiologically stable. **d** Genome-wide CNA profiles showed that after initiation of ICI therapy, CNA changes became less evident in the initial phase of stable disease but increased significantly six months before radiological disease progression. Chromosome regions in shades of red indicate CNA gains, regions in green CNA losses; hGE haploid genome equivalent. Source data are provided as a Source Data file.

## Primary site prediction

For the clinical primary site prediction, patients were independently classified for their putative primary by three experienced oncologists based on clinical and immunohistochemical features as previously described[21], with putative primary tumors being only registered in case of consensus between at least two of the three investigators. Putative primary tumor groups comprised distinct favorable subtypes recognized in the ESMO guidelines[1], namely colon (adenocarcinoma with CK7−, CK20+, CDX2+ immunohistochemistry [IHC] and metastatic spread compatible with colon cancer), head and neck (squamous cell carcinomas with predominant cervical lymph node metastases), breast (adenocarcinoma with predominant axillary lymph node metastases in females) and inner genitals (serous pelvic/peritoneal adenocarcinoma in females), but also upper gastrointestinal tract (adenocarcinoma with leading peritoneal carcinosis and/or abdominal wall infiltration with compatible IHC and without colonic profile), lung (CK7+ adeno- or squamous cell carcinomas with a metastatic pattern suggestive of lung cancer, that is, mediastinal lymph nodes, predominant tumor burden within the chest and distant metastatic sites typical of lung cancers) and anal/cervix (squamous cell carcinomas with pelvic masses and or inguinal lymph node metastases), as well as fully enigmatic cases where no or no unanimous assignment could be made. Primary site prediction was then correlated with treatment response and survival.

## DNA methylation−based CNA profiling of FFPE tissue samples

Methylation analysis was performed in cases where FFPE DNA was available. In brief, samples were analyzed at the microarray unit of the DKFZ Genomics and Proteomics Core Facility using the Illumina Infinium MethylationEPIC BeadChip array according to the manufacturer's instructions. Copy-number variation analysis from the methylation array data was performed as described by Capper et al.[78] using the conumee[79] Bioconductor package v.1.34.0 in R v.4.3.1.

## ccfDNA isolation and sequencing library preparation

ccfDNA from healthy control samples and CUP patients was isolated from plasma (median: 10.2 ml; range: 4.3–14.2 ml) using the QIAamp MinElute ccfDNA Midi Kit (Qiagen) following the manufacturer's instructions and stored at −80 °C. Purified DNA was quantified with Qubit 2.0 using the dsDNA High Sensitivity Kit (Thermo Fisher Scientific). Size distribution of isolated DNA was analyzed with TapeStation 4150 using the cell-free DNA ScreenTape Assay (Agilent Technologies) according to the manufacturer's instructions. For both capture-based targeted and shallow whole-genome sequencing, between 8 and 40 ng of ccfDNA were size-selected for DNA fragments between 90 and 220 bp using SPRIselect (Beckman Coulter) according to the manufacturer's recommendations.

Sequencing libraries were prepared using the KAPA Hyper Prep Kit (Roche Diagnostics International AG) with an optimized manufacturer's protocol and without additional DNA fragmentation. In brief, up to 4 ng (median: 4 ng; range: 1.4–4 ng) of size-selected ccfDNA were subjected to end-repairing, A-tailing and adapter ligation with xGen

Dual Index UMI adapters (Integrated DNA Technologies, Inc). These unique adapters enable superior low-frequency variant detection in low-input samples like ccfDNA and contain both unique molecular identifiers (UMIs) barcoding individual molecules in a sample and unique combined P5 and P7 dual-index sequences that are both unique to a single sample. Following adapter ligation for 15 h at 16 °C to achieve high ligation efficiency, libraries were purified using SPRIselect (Beckman Coulter) according to the manufacturer's protocol. Finally, libraries were amplified in 11-12 PCR cycles and the clean-up of amplified libraries were performed according to the manufacturer's protocol. The size distribution of libraries was confirmed with TapeStation 4150 using the High Sensitivity D1000 ScreenTape Assay (Agilent Technologies) according to the manufacturer's instructions. Finally, the amplified libraries were qPCR-based quantified with adapter-specific primers using the KAPA Library Quantification Kit (Roche Diagnostics International AG) and utilized for both capture-based targeted sequencing and shallow whole-genome sequencing as detailed below.

## Shallow whole-genome sequencing (sWGS) of ccfDNA

Libraries were cleaned up a second time to remove adapter dimers using SPRIselect (Beckman Coulter) according to the manufacturer's recommendations. The libraries were pooled in equimolar amounts and sequenced in multiplexes of 51 sWGS libraries per lane (10 nM multiplex input) using 150 bp paired-end runs on Illumina NovaSeq 6000 S4 v.15 NGS platform (Illumina) at the DKFZ NGS Core Facility according to the manufacturer's instructions. The quality scores Q30 were between 87.6% and 87.2%, total read count prior deduplication was 38 444 522x mean coverage per library (range: 6480 950x – 90,511 187x). Raw sequencing reads were processed and aligned using the automated pipeline OTP[80].

## CheCUP panel design

We designed a customized CheCUP targeted mutation panel using 120 bp long predesigned 5'-biotinylated xGen Lockdown probes (Integrated DNA Technologies, Inc). This CheCUP panel was precisely tailored to our CUP patient cohort targeting 61 patient-specific hotspot mutation regions in 41 different genes (Supplementary Table 5). We selected in mean two hotspot mutations from almost each of the 34 CUP patients showing the highest allele frequencies in the genomic profiling of tumor FFPE samples from metastatic lesions. To ensure high probe specificity, we designed the panel through the xGen Hyb Panel Design Tool (Integrated DNA Technologies, Inc) using genome build hg19 NCBI Build 37.1/GRCh37 as reference. We also assembled the panel so that each base of a target region was covered by two probes (2x tiling design). In total, the panel contained 253 probes (Supplementary Data 2 and 3) spanning only approximately 12 800 bp of the genome (0.0004%), including complete exon coverage of the *TP53* gene. Using this low panel size combined with molecular barcoding enabled superior low-frequency SNP detection even in minute amounts of ctDNA by high-depth targeted sequencing.

## Capture-based targeted sequencing of ccfDNA

Target enrichment was performed according to the xGen hybridization capture of DNA libraries protocol (Integrated DNA Technologies, Inc). In brief, equimolar amounts of six different libraries with unique indices were pooled (3 µg total library input) in a single capture hybridization reaction, and human Cot DNA and xGen Universal Blockers-TS Mix (both Integrated DNA Technologies, Inc) were added as blocking agents. Our CheCUP panel probes were hybridized to the libraries using the xGen Hybridization and Wash Kit (Integrated DNA Technologies, Inc) for 16 h at 65 °C. The capture reaction and extensive washing steps were performed using Dynabeads M-270 Streptavidin beads and xGen Hybridization and Wash Kit, respectively, according to the manufacturer's protocol. Captured libraries were on-beads amplified in 16 PCR cycles with xGen Library Amplification Primers (Integrated DNA Technologies, Inc) complementary to the P5 and P7 sequences of Illumina adapters and KAPA HiFi HotStart ReadyMix (Roche Diagnostics International AG). The post-capture libraries were purified using SPRIselect (Beckman Coulter) and qPCR-based quantified with adapter-specific primers using the KAPA Library Quantification Kit (Roche Diagnostics International AG). Measurement of average fragment length was performed with TapeStation 4150 using the High Sensitivity D1000 ScreenTape Assay (Agilent Technologies) according to the manufacturer's instructions. Always 54 equimolar multiplexed target-enriched libraries (10 nM multiplex input) were then sequenced using 150 bp paired-end runs on Illumina NovaSeq 6000 S4 v.15 NGS platform (Illumina) at the DKFZ NGS Core Facility according to the manufacturer's instructions. The quality scores Q30 were between 90% and 89%, total read count was 44 937 007x mean coverage per library (range: 7 347 738x – 153 278 584x). We achieved 424 535x mean target coverage (median: 412 567x; range: 30 792x – 1 000 042x) prior duplication removal and 1 678x average unique target coverage (median: 1 607x; range: 262x – 4 376x) after deduplication.

## Sequencing analysis of ccfDNA

Raw sequencing reads were processed and aligned using the Subread aligner[81]. PCR duplicates and process artifacts were detected and removed by molecular barcoding using the UMI-tools[82]. Genome-wide copy number profiles and tumor fractions (TFx) were estimated from sWGS data using the ichorCNA algorithm[83] (https://github.com/broadinstitute/ichorCNA) in R (version 4.1.0). First, HMMcopy Suite (http://compbio.bccrc.ca/software/hmmcopy/) was used to partition the genome into equally sized bins of 1 Mb. The read counts were corrected for GC content and mappability biases using the HMMcopy R package. A Bayesian statistical framework of the hidden Markov model (HMM) and an expectation-maximization (EM) algorithm were used to predict CNAs and estimate TFx. A reference panel of normal samples was generated from the sWGS data of the 19 healthy subjects for CNA analysis. The aneuploidy score (AS) was calculated from the genome-wide copy number profiles as the sum total of altered chromosome arms, as described by Taylor et al.[48].

To calculate the ctDNA content, SNVs/indels with a variant allele fraction (VAF) of ≥0.1% were averaged to generate the mean VAF values of each patient. ctDNA concentrations were expressed in haploid genome equivalents (hGE) per mL of plasma (hGE/mL) and calculated by multiplying the mean ctDNA VAF (determined by CheCUP panel sequencing) or the predicted TFx (determined by sWGS) by the concentration of cell-free DNA (cfDNA) (pg/mL of plasma), as determined by Qubit fluorometry. Based on the assumption that most somatic mutations are heterozygous the resulting values were then divided by 3.3 (for mean ctDNA VAF) or 6.6 (for TFx), as each haploid genomic equivalent weighs 3.3 pg, as previously described by Scherer et al., with the expected relationship between mean ctDNA VAF and TFx being 'mean VAF*2=TFx'. If both mean ctDNA VAF and TFx could be determined in a sample, mean ctDNA VAF had priority for the calculation of ctDNA concentrations due to the higher sensitivity of the targeted NGS

approach. For follow-up samples from one patient, ctDNA contents were calculated by consistently using only one of the sequencing approaches.

## Germline mutation detection by whole-exome sequencing (WES)

Peripheral blood was collected from all patients before ICI treatment initiation (baseline). Mononucleated cells were isolated by Ficoll-Biocoll density centrifugation, and genomic DNA was extracted using the QIAamp DNA Mini Kit (Qiagen) following the manufacturer's instructions. DNA was quantified with Qubit 2.0 using the dsDNA High Sensitivity Kit (Thermo Fisher Scientific) and qualified with TapeStation 4150 using the Genomic DNA ScreenTape Assay (Agilent Technologies) according to the manufacturer's instructions. WES libraries were prepared using the SureSelect XT Human All Exon V7 Enrichment Kit (Agilent Technologies). Equimolar multiplexed libraries were then sequenced using 100 bp paired-end runs on Illumina NovaSeq 6000 S1 platform (Illumina) at the DKFZ NGS Core Facility according to the manufacturer's instructions. The quality scores Q30 were between 91.4% and 92.5%, total read count was 103 129 510x mean coverage per library (range: 88 189 108x – 115 652 520x). Raw sequencing reads were processed and aligned using the automated pipeline OTP[64].

## Targeted gene expression profiling

Targeted mRNA expression profiling was conducted on the Nano-String nCounter gene expression platform (NanoString Technologies, Seattle, Washington) using a 770-gene panel (PanCancer Human IO 360 Panel). Immune gene signatures and expression profiling was performed as previously described[45].

## Gene expression data analysis

Analysis of expression data and estimation of the abundance of 14 immune cell populations (B cells, CD45+ cells, CD56dim natural killer (NK) cells, CD8+ T cells, cytotoxic cells, dendritic cells, exhausted CD8+ T cells, macrophages, mast cells, neutrophils, NK cells, T cells, Th1 cells and Treg cells) were carried out as described before[45,46] (Supplementary Data 4). The Spearman correlation between immune cell populations and TMB was calculated, and the significance was assessed using the Benjamini-Hochberg method.

## Statistical analysis

Statistical analyses and data visualization were performed using R version 4.1.0 (www.r-project.org) or GraphPad Prism version 7.03 (GraphPad Software), and a two-sided $p$ value < 0.05 was considered to be statistically significant. All patients enrolled into the trial had valid TMB assessment and received trial medication, i.e., there is only one analysis population for efficacy and safety. The Kaplan-Meier method was used to estimate PFS and OS. Prognostic impact of genomic and clinical baseline parameters on PFS and OS was assessed using log-rank test and Cox regression models. Secondary analyses of the primary endpoint comprised a multivariable Cox regression model including relevant prognostic factors. Fisher's exact test was used to compare response rates. Correlation analyses were performed based on Spearman's correlation coefficient. Group and sample comparisons were made using either two-tailed Mann–Whitney U test or Wilcoxon matched-pairs signed rank test. No adjustment for multiple testing was performed. Incidence and severity of adverse events were analyzed for the safety population. Due to the premature discontinuation of the trial, which resulted in a limited sample size, no confirmatory testing was performed. Instead, all analyses were exploratory and $P$ values are to be interpreted accordingly.

## Reporting summary

Further information on research design is available in the Nature Portfolio Reporting Summary linked to this article.

## Data availability

The raw clinical and imaging data are protected due to patient privacy protected due to patient privacy regulations and are available from the corresponding author upon request for 10 years. Data are located in controlled access data storage at the University Hospital Heidelberg. De-identified clinical data are available upon request and only for research purposes. A first response to access requests will be provided within two weeks of written request. Sharing such data would require approval of the institutional ethics committees. De-identified data will then be transferred to the inquiring investigator over secure data file transfer. Data will then be available for six months. For the CheCUP panel design for targeted ccfDNA sequencing genome build hg19 NCBI Build 37.1/GRCh37 was used as reference. FFPE-DNA, FFPE-RNA and ccfDNA raw sequencing data were deposited into the European Genome-Phenome Archive (EGA) database under Study ID EGAD00001011130. The raw sequencing data are available under controlled access due to privacy policy regulations and in order to ensure that no data are used by for-profit organizations. Data are available upon request to the Data Access Committee (A. Krämer, A. Stenzinger, D. Kazdal, B. N. Kraft; contact: a.kraemer@dkfz.de). The EGA will then create an EGA account with the relevant permissions on our behalf. The processed sequencing data are available within the Source Data file. The study protocol is available as Supplementary Note in the Supplementary Information file. The Informed Consent and the Statistical Analysis Plan are available upon request. All remaining data that support the findings of this study and that are necessary to interpret, verify and extend the research in the article are available within the Article, the Supplementary Information or the Source Data file. Source data are provided with this paper.

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

## Acknowledgements

Funding and investigational drugs were provided by BMS (BMS study ID: CA209-8WY). The sponsor was not involved in the trial design, data collection, analysis or manuscript writing. We thank the NGS Core Facility, German Cancer Research Center (DKFZ), for providing excellent sequencing, Illumina Methylation arrays and preliminary data processing services. We thank Michael Menzel, Institute of Pathology, University of Heidelberg, Heidelberg, Germany for his support regarding the availability of sequencing data. MP was supported by the Physician scientist program of the Medical Faculty of the University of Heidelberg.

## Author contributions
A.K., T.B., and A.S. designed the trial. M.P., M.S., B.K., H.L., U.T.H., G.H., L.W., M.B., T.E., P.S., T.B., and A.K. recruited patients, contributed to patient care and collected the clinical data. M.P., T.H., and A.K. analyzed and interpreted the clinical data. B.N.K. designed the ctDNA analysis and performed the ctDNA sample experiments. T.W. performed the bioinformatic analyses of ctDNA, WES sequencing and methylation data. B.N.K. interpreted the ctDNA and WES sequencing data. A.S., D.K., M.B., and M.K. provided sequencing of tissue samples and central pathology review. K.K. provided the gene expression profiling analysis. S.D. provided central radiology review. T.H. created the statistical plan and made the sample size calculation. B.N.K. and T.H. conducted the statistical analysis. M.P., B.N.K., and A.K. wrote the first draft of the manuscript. All authors participated in developing or revising the manuscript, reviewed, and approved the final draft of the manuscript for publication.

## Funding

## Competing interests
M.S. received consultation fees and honoraria from BMS. B.K. received grants/research support from Roche, Morphosys, MSD and Hexal, and honoraria from Roche. U.T.H. received grants/research support from Celgene and Roche, and honoraria from Roche, Servier, Novartis and Merck Serono. G.H. received consultation fees from Roche and AstraZeneca, and honoraria from Roche, Amgen, Pierre-Fabre, Astra-Zeneca, Alexion, Servier, Octapharma, Abbvie, GSK and Beigene. L.W. received honoraria from Roche and Servier and travel support from Amgen. M.B. received consultation fees from Roche, Incyte, Bayer, MSD and Ipsen. A.S. received grants/research support from Bayer, BMS, Chugai and Incyte, and consultation fees/honoraria from Aignostics, Amgen, Astra Zeneca, Bayer, BMS, Eli Lilly, Illumina, Incyte, Janssen, MSD, Novartis, Pfizer, Qlucore, Roche, Seattle Genetics, Takeda and Thermo Fisher. T.B. has worked as a study oncologist for the CUPISCO trial, which is sponsored by Roche and has received reimbursement for study-related travels as well as remuneration for his work as a study oncologist for the benefit of his employer. A.K. received consultation fees and honoraria from Roche and grants/ research support from BMS and Molecular Health. All remaining authors declare no competing interests.

## Additional information

**Peer review information** : *Nature Communications* thanks Hidetoshi Hayashi, Opeyemi Jegede and the other, anonymous, reviewer(s) for their contribution to the peer review of this work. A peer review file is available.

[1]Clinical Cooperation Unit Molecular Hematology/Oncology, German Cancer Research Center (DKFZ) and Department of Internal Medicine V, University of Heidelberg, Heidelberg, Germany. [2]Department of Internal Medicine V, University of Heidelberg, Heidelberg, Germany. [3]Department of Medical Oncology, National Center for Tumor Diseases (NCT), University of Heidelberg, Heidelberg, Germany. [4]Department of Medical Oncology, Evangelische Kliniken Essen-Mitte, Essen, Germany. [5]Department of Internal Medicine II, Augsburg University Medical Center and Bavarian Cancer Research Center (BZKF), Partner Cite Augsburg, Augsburg, Germany. [6]Department of Internal Medicine III, Marienhospital Stuttgart, Stuttgart, Germany. [7]Department of Medicine II, University Cancer Center Leipzig (UCCL), Leipzig University Medical Center, Leipzig, Germany. [8]Department of Internal Medicine III, Ameos Krankenhausgesellschaft Ostholstein, Eutin, Germany. [9]Department of Internal Medicine, Comprehensive Cancer Center, University of Munich, Munich, Germany. [10]Department of Gastroenterology, Hepatology and Infectiology, University Hospital Tübingen, Tübingen, Germany. [11]Department of Internal Medicine II, Jena University Hospital, Jena, Germany. [12]Onkologische Gemeinschaftspraxis, Gütersloh, Germany. [13]Division of Biostatistics, German Cancer Research Center (DKFZ), Heidelberg, Germany. [14]Division of Radiology, German Cancer Research Center (DKFZ), Heidelberg, Germany. [15]Institute of Pathology, University of Heidelberg, Heidelberg, Germany. [16]Center for Personalized Medicine (ZPM), University of Heidelberg, Heidelberg, Germany. [17]These authors contributed equally: Maria Pouyiourou, Bianca N. Kraft. ✉e-mail: a.kraemer@dkfz.de

