## [Peer Review File · Nature Communications]

Nivolumab and ipilimumab in recurrent or refractory cancer of unknown primary: a phase II trialREVIEWER COMMENTS

Reviewer #1 (Remarks to the Author): with expertise in cancer of unknown primary, immunotherapy

In this study, M Pouyiorou et al showed the results from CheCUP study in which patients with primary unknown cancer (CUP) received nivolumab and ipilimumab therapy after disease progression of chemotherapy including platinum-based regimens. Objective response rate (ORR) in overall population was 16.2%, the corresponding values in TMB-high and TMB-low cohort was 60% and 7.7%, respectively. This is the first report evaluating the clinical efficacy of anti PD-1/L1 antibody and anti CTLA-4 antibody combination in patients with CUP and, therefore, is of importance. However, the study has critical weaknesses in the following aspects.

The study design itself seems questionable. The primary endpoint was PFS and the sample size was calculated to be 194 with a required event count of 191. Due to insufficient patient enrollment, the study termination was decided and thus enrolled only 31 patients which was obviously under power to draw any conclusions. Additionally, such important information, study termination, should be mentioned in the beginning of the Results section.

The number of patients has been further reduced by making TMB assessment mandatory. Whereas, PD-L1 expression was performed on a clinical basis which resulted in data from only 15 patients. As mentioned in the paper, if the biomarker exploration is one of the main aims of this study, data presented here is insufficient. Additional biomarker analysis such as immune profiling should be done on an exploratory basis.

The timing of imaging evaluations every 3 months also reduces the evaluable population by an additional 13 patients whose study treatment was discontinued prior to the first imaging evaluation. Such loss could have been prevented if the evaluation had been performed every 6-8 weeks, as is usually used in clinical trials. This information should be included in the Figure, patient disposition (Fig. 1b).

The ORR of 16% in the entire cohort is similar to that of nivolumab or pembrolizumab monotherapy (Tanizaki et al, Ann Oncol. 2022 ; 33 (2) : 216-226., Raghav KP et al, J

Immunother Cancer 2022; 10(5):e004822.) and survival benefit in the present study is limited. The authors have not discussed what they consider to be the benefits of adding ipilimumab to nivolumab, or in which patient populations benefit from the combination therapy. A more in-depth biomarker analysis and discussion are needed.

As of the nature of unique disease, various treatment strategies have been developed for CUP, with special attention given to primary site prediction and treatment option according to such predicted primary site. Although immunotherapy can be considered a kind of tumor-agnostic approach, the association between primary site prediction and treatment efficacy of nivolumab plus ipilimumab will be of useful information to the readers.

In this study, it is suggested that change of circulating tumor DNA is parallel to those of radiological images. Similar observation has been reported in other solid tumors, and although it is important to note that the same is true for CUP, the timing of ctDNA assessment need to be discussed to call “early detection”. If the authors believe it is the appropriate timing, it should be mentioned or discussed in the Discussion section.

CheCUP study includes patients with performance status (PS) 2, but the effect of immune checkpoint inhibitor on patients with poor PS remains controversial (Khaki et al, JCO Oncol Pract. 2021 Sep;17(9):583-586.). Indeed, three patients with PS2 in this study experienced a short PFS and OS (<3 months and < 6 months, respectively), possibly resulting in the limited efficacy of the combination treatment. Together with the various issues mentioned above, better study design is recommended in the future.

Reviewer #2 (Remarks to the Author): with expertise in cancer of unknown primary, immunotherapy

First and foremost, I commend the authors on trying to answer a critical query by performing a prospective (CheCUP) phase II trial (N = 31) in CUP patients. The study used Nivolumab plus Ipilimumab in patients with refractory unfavorable CUP. The study was aimed to assess the clinical activity of NivoIpi in these patients using PFS as the PEP, stratified for TMB (high vs. low based on a cutoff >12 mut/mb) using TSO500 NGS panel. The trial highlights the accrual challenge with CUP trials (only 45 (23%) of planned 194 enrolled) and then 31 were eligible with 14 (31%) patients excluded due to insufficient tissue and not meeting inclusion criteria. Despite the limitation of sample size, overall, this is a very well

written manuscript and provides some key insights into the value of immunotherapy in CUP.

I do have a few suggestions for the manuscript:

1. Please mention the number of enrolled patients in the abstract, including the fact that the study did not enroll the planned sample size of the study.
2. I would rephrase saying “clear trend” (line 43) for PFS and OS. Knowing that the study did not accrue, all findings are hypothesis generating at best. I would just mention the mPFS and mOS with their 95%CI.
3. Please mention a statement about toxicity in the abstract. Please note that Grade 3 or immune AEs were seen in a third of the population. Also,
4. Please remove the last statement from introduction as this is a result/conclusion statement.
5. On line 133, please add 95%CI of ORR.
6. On line 144 authors mention a patient treated with ivosidenib for activating IDH1 mutation. Was this patient centrally reviewed for being cholangiocarcinoma?
7. On line 155 and 160, please rephrase “strong trend” and “approximating statistical significance” for above state reasons. These are not required to interpret data for what it is and the sample size is limited to support such a statement. I would simply state that that there is a clinically meaningful difference in PFS and OS between these groups.
8. Can the authors provide the ORR by histology?
9. Please provide a detailed description of study limitations in the discussion section including lack of accrual.

Reviewer #3 (Remarks to the Author): with expertise in onco-genomics, ctDNA

The authors present data and analytical insights from a clinical trial of combination immunotherapy in CUP patients. The study provides interesting data on the use of TMB as a biomarker, and also demonstrates how cell-free DNA could be used to stratify patient responses. While it's not the first study of immunotherapy in CUP, previous studies have had small patient cohorts and insights have been mostly non-conclusive. The paper is well structured and written, with informative figures and mostly reasonable conclusions. I only have a few concerns with the manuscript, see below.

Missing sample size in abstract

The study must mention the sample size in the abstract (N=31 patients with NGS data), so readers can more directly evaluate effect sizes and p-values.

Effect of TMB is reduced in multi-variate analysis

“The favorable effect of high TMB was sustained after adjusting for ECOG and disease burden.” Such a strong conclusion is not well supported by their multi-variate analysis. In the table 4 the authors show a highly diminished effect of TMB on PFS (P=0.6) after factoring in ECOG and disease burden authors. The authors should moderate their conclusions in the discussion, discuss why the effect of TMB appear to be reduced in the multi-variate analysis.

Fig 1c,d p-values

Are the p-values of these two analysis identical (p=0.056)?

ccfDNA concentrations

It is intriguing (and confusing) that the authors found the baseline ccfDNA concentration, but not ctDNA concentration, to be associated with PFS and OS. Logically, we would expect the ctDNA concentration to influence patient survival more than ccfDNA concentration? The ctDNA concentrations were calculated by multiplying the mean ctDNA VAF (determined by CheCUP panel sequencing) or the predicted Tfx (determined by sWGS) by the concentration of cell-free DNA (cfDNA) (pg/mL of plasma)? What is the relationship between patient survival and mean ctDNA VAF and predicted Tfx respectively?

Tumor aneuploidy

Tumor aneuploidy is another proposed biomarker of immune checkpoint blockade therapy response (Spurr et al., Nat Genet, 2022; <https://doi.org/10.1038/s41588-022-01235-4>). Since the authors performed copy number profiling on the baseline ctDNA samples and off-target reads from tumor tissues, can they validate if tumor aneuploidy is associated with immunotherapy response in their cohort?

First evidence

“Despite the small sample size, our trial provides first evidence of TMB as an independent biomarker of immunotherapy efficacy in CUP.” – not sure if they can claim this work as the first evidence, how about reference 47?

Dual vs mono therapy

“With regard to combination treatment, dual immune checkpoint blockade with ipilimumab and nivolumab failed to improve efficacy as compared to nivolumab or pembrolizumab monotherapy.” – I don’t think the authors can conclude this as the baseline characteristics of the patients in this trial could be different from the previous trials. A randomized controlled study is needed to reach this conclusion.

Reviewer #4 (Remarks to the Author): with expertise in biostatistics, clinical trial study design

Dr. Maria Pouyiourou and colleagues report primary results from the CheCUP trial as well as very extensive correlative & ctDNA analysis. The study closed early due to slow accrual after enrolling 45 patients. Below are my comments on their manuscript:

- Given the limited effective sample size compared to the original plan (31 vs 194 patients), clinical efficacy results should be solely limited to descriptive analysis with precision estimates (confidence interval) presented and inferences (p-values) removed.
- The operating characteristics of statistical tests are suboptimal when small sample size is a small (see <https://www.nature.com/articles/nrn3475>). This critique is in reference to the PFS, OS, & ORR analysis findings, including those presented in Table 4.
- What is the rationale for the TMB cutoff of 12 mut/Mb compared to the KEYNOTE-158 study with a cutoff of 10 mut/Mb?
- The last sentence in Introduction section appears out of place. It describes study results and is more appropriate in the Results section.

- The claims made in the sentence in line 354-6 and 366-8 of the Discussion section are not supported by the data presented in this study, especially given the limited sample size.

- Analyses veered into correlating ECOG PS, metastasis burden, and number of prior lines of therapy with clinical outcomes thereby deviating from the study objectives stated in the study protocol.

We thank the reviewers for their time and effort to improve and clarify the manuscript. We now have addressed the reviewers' suggestions in full on a point by point basis. Revisions in the manuscript file are highlighted in yellow.

REVIEWER COMMENTS

Reviewer #1, with expertise in cancer of unknown primary, immunotherapy

In this study, M Pouyiourou et al showed the results from CheCUP study in which patients with primary unknown cancer (CUP) received nivolumab and ipilimumab therapy after disease progression of chemotherapy including platinum-based regimens. Objective response rate (ORR) in overall population was 16.2%, the corresponding values in TMB-high and TMB-low cohort was 60% and 7.7%, respectively. This is the first report evaluating the clinical efficacy of anti PD-1/L1 antibody and anti CTLA-4 antibody combination in patients with CUP and, therefore, is of importance. However, the study has critical weaknesses in the following aspects.

1. The study design itself seems questionable. The primary endpoint was PFS and the sample size was calculated to be 194 with a required event count of 191. Due to insufficient patient enrollment, the study termination was decided and thus enrolled only 31 patients which was obviously under power to draw any conclusions. Additionally, such important information, study termination, should be mentioned in the beginning of the Results section.

Indeed, the trial was initially designed to enroll a total of 194 patients and document 191 events in order to detect a hazard ratio of 0.65 for TMB^{high} versus TMB^{low} patients with 80% power at a two-sided significance level of 5%. Between December 2019 and March 2021, a total of 45 patients were screened and 31 patients enrolled into the trial. Due to insufficient recruitment, the trial was then terminated by the sponsor, thereby highlighting once more and similar to earlier attempts the accrual challenge with CUP trials even in a multicenter setting. As a result, the trial was underpowered and all analyses were exploratory. As suggested by the reviewer, the premature trial discontinuation is now mentioned in the abstract and at the beginning of the results section of the manuscript. Also, since all analyses were exploratory due to the limited sample size, we have now clearly stated and described this statistical approach in the results section.

2. The number of patients has been further reduced by making TMB assessment mandatory. Whereas, PD-L1 expression was performed on a clinical basis which

resulted in data from only 15 patients. As mentioned in the paper, if the biomarker exploration is one of the main aims of this study, data presented here is insufficient. Additional biomarker analysis such as immune profiling should be done on an exploratory basis.

TMB assessment was mandatory because patients were prospectively stratified into TMB^{high} and TMB^{low} cohorts, respectively, in order to determine the impact of TMB on response and survival after dual immune checkpoint inhibitor therapy, as TMB is approved by the FDA as a tissue-agnostic predictor of response to pembrolizumab already. Regarding immune profiling as an additional biomarker we have now determined the immune cell composition of the tumor microenvironment by using NanoString nCounter technology-based RNA expression analysis from 13 cases, for whom tumor tissue RNA was available. By this methodology, the abundance of 14 specific immune cell populations (B cells, CD45⁺ cells, CD56^{dim} natural killer (NK) cells, CD8⁺ T cells, cytotoxic cells, dendritic cells, exhausted CD8⁺ T cells, macrophages, mast cells, neutrophils, NK cells, T cells, Th1 cells and Treg cells) was estimated from the mRNA expression profiles as described earlier (Budczies et al., Oncoimmunology 10: e1860586, 2021). Whereas no significant correlation with PFS or OS was found for any of the 14 immune cell populations analyzed, a significant correlation with TMB was observed for neutrophils and regulatory T (Treg) cells, the presence of which has generally been associated with reduced survival and poor response to ICI treatment. As outside the ICI treatment setting higher TMB levels are associated with poorer survival in many cancer types (Valero et al., Nat. Genet. 53: 11-15, 2021), recruitment of neutrophils and Treg cells into TMB^{high} tumors might contribute to this phenomenon. These findings have now been included into the manuscript and are discussed in the Discussion section.

3. The timing of imaging evaluations every 3 months also reduces the evaluable population by an additional 13 patients whose study treatment was discontinued prior to the first imaging evaluation. Such loss could have been prevented if the evaluation had been performed every 6-8 weeks, as is usually used in clinical trials. This information should be included in the Figure, patient disposition (Fig. 1b).

We agree with the reviewer that monitoring disease progression at shorter intervals could have resulted in a more accurate estimation of the response rate. As described in the results section, patients with high disease burden and an aggressive disease course did not benefit from immunotherapy and demonstrated clinical progression before the first response assessment. In fact, thirteen patients discontinued treatment upon clinical progression during the first and additional eight patients during the second treatment cycle. The vast majority of these patients experienced rapid deterioration of their general condition as a result of disease progression and died shortly thereafter before a subsequent therapy line could be initiated. In these cases,

additional imaging studies were strongly encouraged and were performed whenever feasible as part of the routine patient care. However, even in the cases without documented radiological progression, overall and progression-free survival were still evaluable for the purpose of this study. As suggested by the reviewer, we have now adapted Figure 1b in order to contain this information.

4. The ORR of 16% in the entire cohort is similar to that of nivolumab or pembrolizumab monotherapy (Tanizaki et al, Ann Oncol. 2022 ; 33 (2): 216-226., Raghav KP et al, J Immunother Cancer 2022; 10(5):e004822.) and survival benefit in the present study is limited. The authors have not discussed what they consider to be the benefits of adding ipilimumab to nivolumab, or in which patient populations benefit from the combination therapy. A more in-depth biomarker analysis and discussion are needed.

We thank the reviewer for this comment. We have now revised the discussion regarding this point. Since the patient cohorts treated within both monotherapy trials mentioned by the reviewer differed from the CheCUP cohort in terms of ECOG status and previous therapy lines and patient numbers in all three trials are limited, our results cannot offer concluding evidence regarding the patient population that might benefit from combined over mono-immunotherapy. However, it can certainly not be concluded that dual immune checkpoint inhibition is superior to monotherapy for the overall population. This point is now clearly stated in the Discussion section of the manuscript.

5. As of the nature of unique disease, various treatment strategies have been developed for CUP, with special attention given to primary site prediction and treatment option according to such predicted primary site. Although immunotherapy can be considered a kind of tumor-agnostic approach, the association between primary site prediction and treatment efficacy of nivolumab plus ipilimumab will be of useful information to the readers.

We agree with the reviewer that tissue-of-origin prediction and subsequent correlation with treatment response to immune checkpoint inhibitors may be interesting. Several mostly gene expression profiling and PCR-based classifiers have been suggested for primary site prediction in CUP, which require RNA for analysis. In view of the comprehensive biomarker analysis performed in our trial and the scarcity of biopsy material in most CUP cases, sample availability to perform RNA-based tissue of origin prediction was insufficient. However, as described previously, primary site prediction based on clinical, radiological and immuno-histochemical features was feasible (Bochtler et al., Genes Chromosomes Cancer 61: 551-560, 2022). For that, patients were independently classified for their putative primary by three experienced oncologists, with putative primary tumors being only registered in case of

consensus, as has been described earlier (Bochtler et al., Genes Chromosomes Cancer 61: 551-560, 2022). Consensus was reached in 29/31 (93.5%) cases and led to the registration of a putative primary in 16/31 (51.6%) CheCUP patients, while in 13/31 (41.9%) cases no primary tumor could be suggested. The overall response rate was similar for cases with (2/16; 12.5%) and without (2/13; 15.4%) a registered putative primary tumor. Also, median progression-free survival was virtually identical for patients with (2.5 months, 95% CI 2.2-4.3) and without (2.3 months, 95% CI 1.4-NA; log-rank $p=0.561$) a registered putative primary.

6. In this study, it is suggested that change of circulating tumor DNA is parallel to those of radiological images. Similar observation has been reported in other solid tumors, and although it is important to note that the same is true for CUP, the timing of ctDNA assessment need to be discussed to call “early detection”. If the authors believe it is the appropriate timing, it should be mentioned or discussed in the Discussion section.

In the CheCUP trial, for proof of concept serial plasma samples were collected and processed for ctDNA assessment in parallel to imaging studies during combined ipilimumab and nivolumab administration in three-monthly intervals. This is in accordance with the Lung TRACERx study, one of the largest translational research studies evaluating the clonal evolution of non-small cell lung cancer (NSCLC), in which plasma samples for ctDNA analysis are collected preoperatively and at three-monthly intervals during the first two years of follow-up (Abbosh et al., Nature 616: 553-652, 2023; Jamal-Hanjani et al., N. Engl. J. Med. 376: 2109-2121, 2017).

By applying combined targeted NGS and sWGS sequencing, ctDNA was detected in 90% of patients, allowing for ctDNA-based disease monitoring in 11/13 patients with available follow-up samples. In these cases, longitudinal ctDNA analysis after three months of therapy allowed identification of patients who benefited from immunotherapy irrespective of initial radiologic response at the earliest re-assessment performed. We have therefore concluded that ctDNA assessment at the three months timepoint can be used as a surrogate marker of response to immunotherapy in CUP. We concur with the reviewer that further data is needed to determine the optimum timepoint for ctDNA evaluation and whether ctDNA follow-up analysis at an even earlier timepoint will allow for the same conclusion. In this spirit, most recent data on chemo-immunotherapy in patients with metastatic NSCLC suggest that longitudinal ctDNA assessment at and around week 6 of treatment was able already to predict overall survival and could be used to identify patients at high risk of disease recurrence, irrespective of the patient's treatment response as assessed by imaging (Assaf et al., Nat. Med. 29: 859-568, 2023). We have now discussed this point in greater detail in the Discussion section.

7. CheCUP study includes patients with performance status (PS) 2, but the effect of immune checkpoint inhibitor on patients with poor PS remains controversial (Khaki et al, JCO Oncol Pract. 2021 Sep;17(9):583-586.). Indeed, three patients with PS2 in this study experienced a short PFS and OS (<3 months and < 6 months, respectively), possibly resulting in the limited efficacy of the combination treatment. Together with the various issues mentioned above, better study design is recommended in the future.

As pointed out by the reviewer, patients with poor performance status may experience limited benefit from immunotherapy. Indeed, analyses of several retrospective cohorts across different tumor types suggest that ECOG PS ≥ 2 patients have worse response rates, faster progression and shorter survival on treatment with immune checkpoint inhibitors. On the other hand, however, another trial has found that in patients with MSI^{high} cancers immunotherapy is effective regardless of performance status (Pietrantonio et al., Oncologist 25: 803-809, 2020). As a consequence of these conflicting results, which in addition mostly rely on retrospective data, Khaki and coworkers concluded that "further work is needed to identify more reliable predictive biomarkers to improve selection of patients with poor PS who would benefit from ICIs" (Khaki et al., JCO Oncol. Pract. 17: 583-586, 2021). In addition, they state that "the incorporation of patients with poor PS in prospective trials is urgently needed to further understand the role of ICIs in this population". Also, Pietrantonio and coworkers conclude from their work that "future investigations on ICIs in other patient subgroups with ECOG PS >1 and high chance of response" are warranted. For these reasons and especially as TMB, similar to MSI, is of predictive value for checkpoint inhibitor therapy and the CheCUP trial is stratified into TMB^{high} versus TMB^{low} cohorts, we decided to allow for the inclusion of PS2 patients into our trial.

Reviewer #2, with expertise in cancer of unknown primary, immunotherapy

First and foremost, I commend the authors on trying to answer a critical query by performing a prospective (CheCUP) phase II trial (N = 31) in CUP patients. The study used Nivolumab plus Ipilimumab in patients with refractory unfavorable CUP. The study was aimed to assess the clinical activity of Nivolpi in these patients using PFS as the PEP, stratified for TMB (high vs. low based on a cutoff >12 mut/mb) using TSO500 NGS panel. The trial highlights the accrual challenge with CUP trials (only 45 (23%) of planned 194 enrolled) and then 31 were eligible with 14 (31%) patients excluded due to insufficient tissue and not meeting inclusion criteria. Despite the limitation of sample size, overall, this is a very well written manuscript and provides some key insights into the value of immunotherapy in CUP. I do have a few suggestions for the manuscript:

1. Please mention the number of enrolled patients in the abstract, including the fact that the study did not enroll the planned sample size of the study.

The abstract has been revised accordingly.

2. I would rephrase saying “clear trend” (line 43) for PFS and OS. Knowing that the study did not accrue, all findings are hypothesis generating at best. I would just mention the mPFS and mOS with their 95%CI.

The statement was rephrased according to the reviewer’s suggestion.

3. Please mention a statement about toxicity in the abstract. Please note that Grade 3 or immune AEs were seen in a third of the population.

We have now included information about toxicity in the abstract.

4. Please remove the last statement from introduction as this is a result/conclusion statement.

The statement has been removed as suggested by the reviewer.

5. On line 133, please add 95%CI of ORR.

We have now added the 95% CI of ORR.

6. On line 144 authors mention a patient treated with ivosidenib for activating IDH1 mutation. Was this patient centrally reviewed for being cholangiocarcinoma?

We thank the reviewer for this comment. Histology and immunohistochemistry in this case was compatible with a primary in the upper gastrointestinal tract (CK7+, CK20-) but not conclusive for cholangiocarcinoma. Reference radiology at study eligibility screening did not confirm the diagnosis of cholangiocarcinoma according to the differential diagnostic algorithms of the ESMO CUP guidelines. We have now clarified this point in the manuscript.

7. On line 155 and 160, please rephrase “strong trend” and “approximating statistical significance” for above state reasons. These are not required to interpret data for what it is and the sample size is limited to support such a statement. I would simply state that there is a clinically meaningful difference in PFS and OS between these groups.

We have now revised the paragraph according to the reviewer’s suggestions.

8. Can the authors provide the ORR by histology?

We have now provided detailed information on ORR by histology in Supplementary Table 3.

	Total	Adeno-carcinoma	Squamous cell carcinoma	Poorly differentiated carcinoma	Carcinoma with sarcomatoid differentiation
Patients (n)	31	20	5	4	2
Overall response n (%)	5 (16.1)	2 (10)	1 (20)	1 (25)	1 (50)
Best overall response					
Complete response	2 (6.5)	1 (5)	1 (20)	0	0
Partial response	3 (9.7)	1 (5)	0	1 (25)	1 (50)
Stable disease	1 (3.2)	1 (5)	0	0	0
Progressive disease	12 (38.7)	7 (35)	3 (60)	2 (50)	0
Early study discontinuation	13 (41.9)	10 (50)	1 (20)	1 (25)	1 (50)

9. Please provide a detailed description of study limitations in the discussion section including lack of accrual.

We have now included a description of the main limitations of the trial in the Discussion section of the manuscript.

Reviewer #3, with expertise in onco-genomics, ctDNA

The authors present data and analytical insights from a clinical trial of combination immunotherapy in CUP patients. The study provides interesting data on the use of TMB as a biomarker, and also demonstrates how cell-free DNA could be used to stratify patient responses. While it's not the first study of immunotherapy in CUP, previous studies have had small patient cohorts and insights have been mostly non-conclusive. The paper is well structured and written, with informative figures and mostly reasonable conclusions. I only have a few concerns with the manuscript, see below.

1. Missing sample size in abstract. The study must mention the sample size in the abstract (N=31 patients with NGS data), so readers can more directly evaluate effect sizes and p-values.

The abstract has been revised accordingly.

2. Effect of TMB is reduced in multi-variate analysis. The favorable effect of high TMB was sustained after adjusting for ECOG and disease burden.” Such a strong conclusion is not well supported by their multi-variate analysis. In the table 4 the authors show a highly diminished effect of TMB on PFS (P=0.6) after factoring in ECOG and disease burden authors. The authors should moderate their conclusions in the discussion, discuss why the effect of TMB appear to be reduced in the multi-variate analysis.

As suggested by the reviewer, we have now revised the discussion on multivariate analysis in order to clarify the results and their interpretation. Although a favorable effect of TMB^{high}, particularly on OS (univariate HR 0.32, multivariate HR 0.45) was sustained to some extent in the multivariate analysis, poor ECOG and high disease burden seem to diminish the effect of TMB especially on PFS, probably due to the fact that both lower ECOG and lower disease burden were more frequent in the TMB^{high} group.

3. Fig 1c,d p-values. Are the p-values of these two analysis identical (p=0.056)?

Yes. We confirm that they are.

4. ccfDNA concentrations. It is intriguing (and confusing) that the authors found the baseline ccfDNA concentration, but not ctDNA concentration, to be associated with PFS and OS. Logically, we would expect the ctDNA concentration to influence patient survival more than ccfDNA concentration? The ctDNA concentrations were calculated by multiplying the mean ctDNA VAF (determined by CheCUP panel sequencing) or the predicted Tfx (determined by sWGS) by the concentration of cell-free DNA (cfDNA) (pg/mL of plasma)? What is the relationship between patient survival and mean ctDNA VAF and predicted Tfx respectively?

Indeed, we expressed the ctDNA content in haploid genome equivalents (hGE) per mL of plasma and calculated it by multiplying the mean VAF or the predicted Tfx by the concentration of ccfDNA and then dividing by 3.3, using the assumption that each haploid genome equivalent weighs 3.3 pg as previously described by Scherer and coworkers (Sci. Transl. Med. 8: 364ra155, 2016). To us, reporting ctDNA levels as mean VAF or predicted Tfx seemed inaccurate because this only reflects ctDNA levels as a percentage of ccfDNA content.

The release of ccfDNA is not only influenced by tumor features (e.g. tumor site, disease burden, tumor microenvironment, rates of proliferation and cell death), but also by other patient-specific factors including physiological conditions (e.g. age, oxidative stress, diet, strenuous exercises), inflammation as well as additional acute and chronic medical conditions (e.g. trauma, surgery, autoimmune diseases, myocardial infarction, sepsis). Also, the total ccfDNA content may be affected by side effects (e.g. immune-related adverse events) of immunotherapy as well. Nevertheless, we have compared patient survival with mean VAF and predicted Tfx, respectively, as suggested by the reviewer. However, neither mean VAF nor predicted Tfx were associated with PFS and OS:

As mentioned earlier, nearly half of the CheCUP patients rapidly deteriorated, discontinued treatment upon clinical progression during the first treatment cycle and died shortly thereafter. Their poor general condition was reflected by highly elevated ccfDNA levels, thereby explaining why the baseline ccfDNA concentrations were associated with poor PFS and OS. On the contrary, mean VAF and predicted TFx of these patients were low compared with other patients, as their total ccfDNA did not merely reflect increased ctDNA levels due to disease progression. The lack of association between ctDNA content - expressed in hGE/ml plasma – and survival, on the other hand, may be a consequence of the limited sample size of our study or, more likely, due to amplification and wider spread of ccfDNA levels by deterioration of the patient’s general conditions over increased ctDNA levels by tumor progression only. Studies in other solid tumors have shown that ctDNA can help to predict response to immunotherapies. Similar to our results, several of these studies found that on-treatment ctDNA kinetics rather than baseline pre-treatment ctDNA levels predict for clinical benefit and overall survival (Jensen et al., *Mol. Cancer Ther.* 18: 448-458, 2019; Weiss et al., *Clin. Cancer Res.* 23: 5074-5081, 2017; Ricciuti et al., *J. Immunother. Cancer* 9: e001504, 2021; Kato et al., *Oncoimmunology* 11: 2052410, 2022).

5. Tumor aneuploidy. Tumor aneuploidy is another proposed biomarker of immune checkpoint blockade therapy response (Spurr et al., *Nat Genet*, 2022; <https://doi.org/10.1038/s41588-022-01235-4>). Since the authors performed copy number profiling on the baseline ctDNA samples and off-target reads from tumor

tissues, can they validate if tumor aneuploidy is associated with immunotherapy response in their cohort?

We thank the reviewer for this comment and suggestion, and have now analyzed the impact of tumor aneuploidy on immunotherapy in the CheCUP patient population. Of note, a higher aneuploidy score (Spurr et al., Nat. Genet. 54: 1782-1785, 2022; Taylor et al., Cancer Cell 33: 676-689, 2018) is associated with a significantly shorter progression-free survival both in the total (upper panel) and the TMB^{low} population (lower panel), when aneuploidy is calculated from tumor tissue. Moreover, 4 of 11 patients (36.4%) with low aneuploidy score, but none of eight patients with high aneuploidy score achieved an objective response.

On the other hand, no association between survival and aneuploidy was found when the aneuploidy score was calculated from ctDNA, because in some patients low baseline ctDNA levels prevented effective DNA copy number profiling by sWGS. These data have now been included into the revised version of the manuscript.

6. First evidence. "Despite the small sample size, our trial provides first evidence of TMB as an independent biomarker of immunotherapy efficacy in CUP." – not sure if they can claim this work as the first evidence, how about reference 47?

We have now clarified this point in the Discussion section. As pointed out by the reviewer, initial data on the effect of TMB on immunotherapy efficacy in CUP has indeed been reported before (Tanizaki et al., Ann. Oncol. 33: 216-226, 2022). However, the respective analysis has only been performed on a retrospective basis. Our trial is the first to prospectively analyze the effect of high TMB in this context.

7. Dual vs mono therapy. "With regard to combination treatment, dual immune checkpoint blockade with ipilimumab and nivolumab failed to improve efficacy as compared to nivolumab or pembrolizumab monotherapy." – I don't think the authors can conclude this as the baseline characteristics of the patients in this trial could be different from the previous trials. A randomized controlled study is needed to reach this conclusion.

We have now revised this point accordingly. See also comment 4 to reviewer #1.

Reviewer #4, with expertise in biostatistics, clinical trial study design

Dr. Maria Pouyiourou and colleagues report primary results from the CheCUP trial as well as very extensive correlative & ctDNA analysis. The study closed early due to slow accrual after enrolling 45 patients. Below are my comments on their manuscript:

1. Given the limited effective sample size compared to the original plan (31 vs 194 patients), clinical efficacy results should be solely limited to descriptive analysis with precision estimates (confidence interval) presented and inferences (p-values) removed.

We agree with the reviewer and have now amended the results section accordingly by stating that due to the limited sample size and the premature trial discontinuation all analyses were exploratory. P-values are reported and interpreted as descriptive rather than confirmatory values.

2. The operating characteristics of statistical tests are suboptimal when small sample size is a small (see <https://www.nature.com/articles/nrn3475>). This critique is in reference to the PFS, OS, & ORR analysis findings, including those presented in Table 4.

We agree with the reviewer that the small sample size is why no formal confirmatory claims can be made. Particularly, the events per variable (EVP) ratio in the multivariate analysis (Table 4) is quite low due to the small sample size. Besides a low EVP ratio, omitting true predictors from the model can introduce bias and both sources of potential bias need to be balanced in our situation. There is some evidence that omitting important predictors dominates bias (Steyerberg et al., J. Clin. Epidemiol. 64: 1464-1465, 2011), which is why we decided on this rather large model. We have now added the low EVP ratio as additional limitation for the multivariate analysis.

3. What is the rationale for the TMB cutoff of 12 mut/Mb compared to the KEYNOTE-158 study with a cutoff of 10 mut/Mb?

We thank the reviewer for the important remark. Before the trial was initiated, the authors led and were involved in several international consortia analyzing the technical details of TMB estimation based on tumor-only panel sequencing. Collectively, the resulting studies demonstrated significant variance of TMB scores depending on several parameters, particularly around the cut point of 10 mutations/Mb (Sha et al., Cancer Discov. 10: 1808-1825, 2020; Allgäuer et al., Transl. Lung Cancer Res. 7: 703-715, 2018; Budczies et al., Ann. Oncol. 30: 1496-1506, 2019; Buchhalter et al., Int. J. Cancer 144: 848-858, 2019; Vega et al., Ann. Oncol. 32: 1626-1636, 2021; Budczies et al., Lung Cancer. 142: 114-119, 2020; Kazdal et al., J. Thorac. Oncol. 14: 1935-1947, 2019; Endris et al., Int. J. Cancer 144: 2303-2312, 2019; Stenzinger et al., J. Thorac. Oncol. 15: 1177-1189, 2020; Merino et al., J. Immunother. Cancer 8: e000147, 2020). This led to the proposal of a three-tier classification system, which accounts for the observed multifactorial variability. Our bridging studies using TMB determined by whole exome sequencing (WES; as initially demonstrated by Rizvi et al., Science, 2015) revealed that a WES-bridged value of 10 mutations/Mb when detected by the TSO500 panel was associated with a confidence interval from approx. 8 to 11 mutations/Mb. Therefore, we decided to employ a slightly higher cut point of 12 mutations/Mb to support the enrichment of TMB^{high} patients in the CheCUP trial.

We have additionally performed a retrospective assessment to identify the optimum TMB cutoff for our study cohort. Using the resulting optimal cutoff of 8.63 mutations/Mb resulted in the same cohort composition, since all patients with TMB > 12 were also TMB > 8,63 (adjusted p-values: OS p=0.18, PFS p=0.17).

4. The last sentence in Introduction section appears out of place. It describes study results and is more appropriate in the Results section.

This sentence has now been deleted from the revised version of the manuscript.

5. The claims made in the sentence in line 354-6 and 366-8 of the Discussion section are not supported by the data presented in this study, especially given the limited sample size.

We have now discussed the limitations of the small sample size and adjusted both statements.

6. Analyses veered into correlating ECOG PS, metastasis burden, and number of prior lines of therapy with clinical outcomes thereby deviating from the study objectives stated in the study protocol.

Indeed, the analyses mentioned by the reviewer were not predefined in the study protocol. However, baseline characteristics including ECOG performance status, metastasis burden, and number of prior lines of therapy were prespecified in the statistical analysis plan (SAP, paragraph 5.2, page 10) accompanying the study protocol, to be reported and analyzed. We have therefore added exploratory analyses on these parameters after completion of the trial.

REVIEWER COMMENTS

Reviewer #1 (Remarks to the Author):

The authors have responded to the reviewers' suggestions and the paper has been improved. Most reviewers pointed out similar issues, i.e., that the data are exploratory results from the trials that was terminated early, and the inadequacies of the biomarker analysis. Although the sample size is small, this study is the first to demonstrate the efficacy of IO combination therapy in cancers of unknown primary. Additional data on immunological analysis and primary site estimation will provide background data for planning future trials.

Reviewer #2 (Remarks to the Author):

The authors have addressed my reviews and concerns.

Reviewer #3 (Remarks to the Author):

The authors have addressed most of my major concerns in their revised manuscript. See below for my remaining issues:

1. I appreciate that the authors commented on the reduced effect of TMB in their multivariate survival analysis in their revised manuscript. However, the current statement omits that associations are not significant ($p=0.26$, $p=0.6$). Please moderate this statement further and underline that TMB associations are not significant after adjusting for ECOG and metastatic burden:

“Upon adjusting for ECOG and metastasis burden score, the 183 favorable effect of TMB on OS only slightly decreased (HR 0.45, 95% CI 0.11-1.79; $p=0.26$), 184 while the effect on PFS was attenuated again (HR 0.70, 95% CI 0.18-2.67; $p=0.60$; Table 4).”

2. Fig 5,c - Not clear what is the individual contributions from ctDNA fractions and cfDNA plasma levels:

Since the authors compute aggregated ctDNA plasma levels (ctDNA_fraction * qubit_cfDNA_plasma_fraction), the authors should also provide data for the contribution from ctDNA_fractions (VAFs and ichorCNA TF) and cfDNA plasma fractions (Qubit) individually, for example by generating supplementary figures equivalent to fig5,c for each of these individual variables. This is especially important since their previous analysis indicated significant correlation of survival with cfDNA plasma fractions, but not with the aggregated ctDNA levels.

Data presented in Suppl. Fig. 15 is confusing:

It is not clear what is causing the difference between figure S15b,c. Both figures plot TFx vs. VAF, with fig b) showing the raw values, and figure c) showing values after applying the same transformation (correcting for cfDNA concentration in plasma)? For example, there are multiple points in b) with high TFx values and near-zero VAF, why are these missing from plot c)?

Calculation of ctDNA levels: First, it is not clear how the authors compute ctDNA levels in samples that have BOTH VAFs and TFx values, using mean or does VAF take precedence? Second, one would assume that most somatic mutations are heterozygous, so the expected relationship between VAF_mean and TF (ichorCNA) would be: $VAF_mean * 2 = TF$. The authors should consider this point for their ctDNA level calculations.

Reviewer #4 (Remarks to the Author):

Because of poor clinical outcomes and unestablished standard of care beyond first-line treatment among patients with unfavorable CUP, the main merit of the current manuscript is that it provides important data to inform clinical care in this patient population.

There is a big discrepancy between observed and projected patient accrual, and this study only closed after enrolling 31 patients (in 15 months) out of the planned enrollment of 194 patients (in 36 months) -- the assumptions made during the study design were not optimal.

The study team made considerable improvement to the manuscript based on (my and other) initial reviewer comments. They've amply improved conclusions and inferences drawn from this limited sample size study. They performed additional correlative analysis which will, hopefully, provide historical data and insights to design a more practical study in this patient population.

Here a final issue that should be addressed:

1)The latter part of the sentence "Due to the limited sample size, all analyses were exploratory and P values are reported and interpreted descriptively." is not meaningful. P-values are not 'interpreted descriptively', they're obtained from inferential analysis and (science) consumers will interpret the results as such. If p-values were indeed interpreted descriptively, then statements regarding 'statistically significant difference' or lack thereof should be reworded, for example, in lines 177, 192, 247.

Reviewer #1 (Remarks to the Author):

The authors have responded to the reviewers' suggestions and the paper has been improved. Most reviewers pointed out similar issues, i.e., that the data are exploratory results from the trials that was terminated early, and the inadequacies of the biomarker analysis. Although the sample size is small, this study is the first to demonstrate the efficacy of IO combination therapy in cancers of unknown primary. Additional data on immunological analysis and primary site estimation will provide background data for planning future trials.

We thank the reviewer for the positive response.

Reviewer #2 (Remarks to the Author):

The authors have addressed my reviews and concerns.

We thank the reviewer for the positive response.

Reviewer #3 (Remarks to the Author):

The authors have addressed most of my major concerns in their revised manuscript. See below for my remaining issues:

1. I appreciate that the authors commented on the reduced effect of TMB in their multi-variate survival analysis in their revised manuscript. However, the current statement omits that associations are not significant ($p=0.26$, $p=0.6$). Please moderate this statement further and underline that TMB associations are not significant after adjusting for ECOG and met burden: "Upon adjusting for ECOG and metastasis burden score, the 183 favorable effect of TMB on OS only slightly decreased (HR 0.45, 95% CI 0.11-1.79; $p=0.26$), 184 while the effect on PFS was attenuated again (HR 0.70, 95% CI 0.18-2.67; $p=0.60$; Table 4)."

To moderate the statement, the following sentence has been added as suggested by the reviewer: "After multivariate analysis, the effect of TMB on OS ($p=0.26$) and PFS ($p=0.60$) was not statistically significant".

*# 2. Fig 5,c - Not clear what is the individual contributions from ctDNA fractions and cfDNA plasma levels: Since the authors compute aggregated ctDNA plasma levels (ctDNA_fraction * qubit_cfDNA_plasma_fraction), the authors should also provide data for the contribution from ctDNA_fractions (VAFs and ichorCNA TF) and cfDNA plasma fractions (Qubit) individually, for example by generating supplementary figures equivalent to fig5,c for each of these individual variables. This is especially important since their previous analysis indicated significant correlation of survival with cfDNA plasma fractions, but not with the aggregated ctDNA levels.*

As suggested by the reviewer we have now added an additional supplementary figure (Supplementary Fig. 16) that, equivalent to Fig. 5c, provides data on the individual contributions from ctDNA fractions as detected by targeted NGS (VAF, Supplementary Fig. 16a) and sWGS (TFx, Supplementary Fig. 16b), respectively, as well as from ccfDNA levels (Supplementary Fig. 16c).

Data presented in Suppl. Fig. 15 is confusing: It is not clear what is causing the difference between figure S15b,c. Both figures plot TFx vs. VAF, with fig b) showing the raw values, and figure c) showing values after applying the same transformation (correcting for cfdna concentration in plasma)? For example, there are multiple points in b) with high TFx values and near-zero VAF, why are these missing from plot c)?

It is the purpose of the diagrams in Supplementary Fig. 15b,c to demonstrate that – in samples in which both were detectable – mean SNV VAFs and TFx estimates (Supplementary Fig. 15b) as well as the respective ctDNA amounts in hGE/ml plasma derived by these two methods (Supplementary Fig. 15c) correlate with each other. The grey dots in Supplementary Fig. 15b depict samples for which either no TFx values or no SNVs/indels were detectable. Accordingly, one of the values for those samples could not be determined and was set to zero (in order to show that even in cases with high TFx values the alternative method (targeted NGS) sometimes does not provide data). For this reason, these dots do not appear in Supplementary Fig. 15c, as for display purposes the axis scales here are logarithmic and zero values can therefore not be included. The figure legend has been revised in order to better clarify this point.

*# Calculation of ctDNA levels: First, it is not clear how the authors compute ctDNA levels in samples that have BOTH VAFs and TFx values, using mean or does VAF take precedence? Second, one would assume that most somatic mutations are heterozygous, so the expected relationship between VAF_mean and TF (ichorCNA) would be: $VAF_mean * 2 = TF$. The authors should consider this point for their ctDNA level calculations.*

If both mean ctDNA VAF and TFx could be determined in a sample, mean ctDNA VAF had priority for the calculation of ctDNA concentrations due to the higher sensitivity of the targeted NGS approach. For follow-up samples from one patient, ctDNA contents were calculated by consistently using only one of the sequencing approaches. This explanation has now been added to the methods section (page 22) of the manuscript.

We agree with the reviewer that expected relationship between mean VAFs and TFx is $mean\ VAF * 2 = TFx$, as indeed likely most somatic mutations are heterozygous. We have therefore amended the respective methods section (page 22) accordingly: “ctDNA concentrations were expressed in haploid genome equivalents (hGE) per mL of plasma

(hGE/mL) and calculated by multiplying the mean ctDNA VAF (determined by CheCUP panel sequencing) or the predicted TFX (determined by sWGS) by the concentration of cell-free DNA (cfDNA) (pg/mL of plasma), as determined by Qubit fluorometry. Based on the assumption that most somatic mutations are heterozygous the resulting values were then divided by 3.3 (for mean ctDNA VAF) or 6.6 (for TFX), as each haploid genomic equivalent weighs 3.3 pg, as previously described by Scherer et al., with the expected relationship between mean ctDNA VAF and TFX being 'mean VAF*2=TFX' ”.

Reviewer #4 (Remarks to the Author):

Because of poor clinical outcomes and unestablished standard of care beyond first-line treatment among patients with unfavorable CUP, the main merit of the current manuscript is that it provides important data to inform clinical care in this patient population. There is a big discrepancy between observed and projected patient accrual, and this study only closed after enrolling 31 patients (in 15 months) out of the planned enrollment of 194 patients (in 36 months) -- the assumptions made during the study design were not optimal. The study team made considerable improvement to the manuscript based on (my and other) initial reviewer comments. They've amply improved conclusions and inferences drawn from this limited sample size study. They performed additional correlative analysis which will, hopefully, provide historical data and insights to design a more practical study in this patient population. Here a final issue that should be addressed:

1)The latter part of the sentence “Due to the limited sample size, all analyses were exploratory and P values are reported and interpreted descriptively.” is not meaningful. P-values are not ‘interpreted descriptively’, they’re obtained from inferential analysis and (science) consumers will interpret the results as such. If p-values were indeed interpreted descriptively, then statements regarding ‘statistically significant difference’ or lack thereof should be reworded, for example, in lines 177, 192, 247.

We initially added the sentence to emphasize that due to the premature stop of the trial no confirmative analysis/testing was done and all analyses were purely exploratory, and that p-values need to be interpreted as such. We now rephrased the sentence to avoid the misconception pointed out by the reviewer: *“Due to the limited sample size, no confirmatory testing was performed. Instead, all analyses were exploratory and p-values are to be interpreted accordingly”.*

REVIEWERS' COMMENTS

Reviewer #3 (Remarks to the Author):

The authors have addressed all my remaining points.